



# Fusing MODIS and AVHRR products to generate a global 1-km continuous NDVI time series covering four decades

Xiaobin Guan[1,2], Huanfeng Shen[1,3,*], Yuchen Wang[1], Dong Chu[1], Xinghua Li[4], Linwei Yue[5], Xinxin Liu[6], Liangpei Zhang[3,7]

[1]School of Resource and Environmental Sciences, Wuhan University, Wuhan, 430079, PR China.
[2]Department of Geography and Planning, University of Toronto, Toronto M5S3G3, Canada.
[3]Collaborative Innovation Centre of Geospatial Technology, Wuhan 430079, PR China.
[4]School of Remote Sensing and Information Engineering, Wuhan University, Wuhan 430079, PR China.
[5]School of Geography and Information Engineering, China University of Geosciences, Wuhan 430074, PR China.
[6]College of Electrical and Information Engineering, Hunan University, Changsha 410205, PR China.
[7]The State Key Laboratory of Information Engineering in Surveying, Mapping and Remote Sensing, Wuhan University, Wuhan 430079, PR China.

*Correspondence to*: Huanfeng Shen (shenhf@whu.edu.cn)

**Abstract.** Satellite normalized difference vegetation index (NDVI) time-series data are an essential data source for numerous ecological and environmental applications. Although various long-term global NDVI products have been produced with different characteristics over the past decades, there is still an apparent trade-off between the spatiotemporal resolution and time coverage. The Advanced Very High-Resolution Radiometer (AVHRR) instrument can provide the only continuous time series with the longest time coverage since the early 1980s, but with the drawback of a coarse spatial resolution and poor data quality compared to the observations of later instruments. To address this issue, a spatio-temporal fusion-based long-term NDVI product (STFLNDVI) since 1982 was generated in this study, with a 1-km spatial resolution and a monthly temporal resolution. A multi-step processing fusion framework was employed to combine the superior characteristics of Moderate Resolution Imaging Spectroradiometer (MODIS) and AVHRR products, respectively. Simulated and real-data assessments both confirm the ideal accuracy of the fusion result with regard to the spatial distribution and temporal variation. Only a few relatively unsatisfactory results are found due to the poor relationship between the original AVHRR and MODIS data. The evaluations also show that the proposed fusion framework can obtain stable results similar to MODIS data in different years and seasons, even when the temporal distance between the fusion data and the reference data is large. We believe that the STFLNDVI product will be of great significance to characterize the spatial patterns and long-term variations of global vegetation. The NDVI product is available at DOI: http://doi.org/10.5281/zenodo.4734593 (Guan et al., 2021).

## 1 Introduction

Long-term and continuous vegetation records are critical for documenting the changes of terrestrial ecosystems (Chen et al., 2019; Piao et al., 2014; Zhu et al., 2016). Satellite spectral vegetation indices (VI) are important data sources for global





vegetation change studies as they can be used to efficiently monitor the spatial distribution and temporal dynamic of large-scale vegetation (Piao et al., 2009; Piao et al., 2014; Wang et al., 2003; Yang et al., 2013). The normalized difference
vegetation index (NDVI), simply calculated by the visible and near-infrared spectral reflectance, is the most widely used satellite VI. NDVI can not only reflect the vegetation coverage and growth status, but is also closely correlated with the leaf area index (LAI), chlorophyll abundance, biomass, vegetation productivity, and other biophysical parameters (Guan et al., 2017; He et al., 2017; Liu et al., 2012; Lunetta et al., 2006). In the context of dramatic global climate change, obtaining long-term remote-sensed NDVI time series is of great importance to advance the knowledge of historical vegetation variation and
its interaction with global change.

**Table 1. Characteristics of the global coarse-resolution NDVI instruments with more than 10 years of coverage.**

| Instrument | Satellite | Spatial resolution | Time coverage |
|---|---|---|---|
| AVHRR 2 | NOAA-7, 9, 11, 14 | 1.1 km, 0.05°, 1/12° | 1981–2001 |
| AVHRR 3 | NOAA-16, 17, 18, 19 MetOp-A, B, C | 1.1 km, 0.05°, 1/12° | 2000 to present |
| SeaWiFS | OrbView-2 | 4 km | 1997–2010 |
| SPOT-Vegetation | SPOT 4 and 5 | 1 km | 1998–2013 |
| MODIS | Terra, Aqua | 250 m to 1 km, 0.05° | 2000 to present |
| TM/ETM+/OLI | Landsat 4, 5, 7, 8 | 30 m | 1972 to present |

In the past decades, a series of satellite sensor systems have been launched into space, which now provide various NDVI
products at different spatial and temporal resolutions (Brown et al., 2006; Fensholt and Proud, 2012; Miura et al., 2012; Tarnavsky et al., 2008). With the accumulation of historical satellite observations, it is now possible to undertake long-term (>10 years) investigation into global or regional vegetation change (Cao et al., 2018). Among the different systems, as shown in Table 1, data from the Advanced Very High-Resolution Radiometer (AVHRR), the Moderate Resolution Imaging Spectroradiometer (MODIS), and the Satellite Pour l'Observation de la Terre (SPOT) can be used to form long-term
continuous dataset products with high consistency, which have been widely applied in studies of the global carbon cycle and climate change at different scales (He et al., 2017; Nayak et al., 2016; Zhang et al., 2020; Zhang et al., 2017a). Although the Landsat series of satellites can offer a finer resolution and very long time span, it is limited by the long revisit period of 16 days and thus difficult to obtain a valid continuous time series with higher frequency for long-term vegetation monitoring at large scales (Shen et al., 2016). The Sentinel-2 satellites can acquire data both with satisfactory spatial and temporal
resolutions, but they can only provide records since 2015 (Drusch et al., 2012). Some other systems can also provide data that can be used to generate coarse-resolution global NDVI continuous time series, such as the Project for On-Board Autonomy (PROBA) Vegetation instrument, Sea-Viewing Wide Field-of-view Sensor (SeaWiFS), and the Visible Infrared Imaging Radiometer Suite (VIIRS) instrument on the Suomi National Polar-orbiting Partnership (NPP), but their limited time coverage can not support decades vegetation monitoring (Pinzon and Tucker, 2014).



The AVHRR sensors mounted on the National Oceanic and Atmospheric Administration (NOAA) satellites are the only series of instruments that can provide the longest NDVI records with a continuous time series from 1981 (Eidenshink and Faundeen, 1994; Fensholt et al., 2006; Goward et al., 1993). There have been two generations of instruments onboard the NOAA-7 through MetOp-C satellites, and the early generation of AVHRR/2 instrument was replaced with the AVHRR/3 instrument since November 2000. Although the AVHRR sensors were not initially designed for vegetation study, their

ability to capture reflectance in the visible (0.58–0.68 µm) and near-infrared (0.725–1.10 µm) range means that they do have potential in this regard. The original spatial resolution of the AVHRR instrument was 1.1 km at nadir, but there was great distortion in the edges because of the wide swath width of approximately 2500 km. Due to the huge volume of AVHRR data and the limited storage capacity in the 1980s and 1990s, the global area coverage (GAC) data were partially resampled onboard by retaining only one line out of every three, and were finally stored at a resolution of approximately 4 km (Pinzon

and Tucker, 2014). Only the limited Local Area Coverage (LAC) and High Resolution Picture Transmission (HRPT) data have been stored at the raw resolution since the early 1990s, but these data are usually only available for specific areas of the world (Eidenshink and Faundeen, 1994). With the launch of the MetOp-A satellite, the entire global coverage of 1.1-km resolution data has been provided since late October 2006 by the Full Resolution Area Coverage (FRAC) data. As a result, in order to acquire a continuous long-term time series beginning in the 1980s, successive efforts have been conducted to

produce AVHRR NDVI products based on the GAC record. Examples are the third-generation Global Inventory Modeling and Mapping Studies (GIMMS 3g) product (Pinzon and Tucker, 2014; Tucker et al., 2005); the Land Long Term Data Record (LTDR) project (Pedelty et al., 2007); the Pathfinder AVHRR Land (PAL I and II) datasets (Gutman and Ignatov, 1996); and the Fourier-Adjusted Sensor and Solar zenith angle corrected, Interpolated Reconstructed (FASIR) adjusted NDVI (Los et al., 2000). Among these examples, unsuccessfully corrected artifacts in the PAL and LTDR datasets have been

reported, and the GIMMS 3g dataset is the only updated global coverage AVHRR NDVI dataset covering the full period from 1981 to present with the highest possible data quality (Fensholt and Proud, 2012; Tian et al., 2015). Due to the advantage of the long time coverage and the limitation of coarse spatial resolution, the GIMMS 3g product has usually been employed to study decades-long change in vegetation at a large scale (Peng et al., 2013; Piao et al., 2009; Xiao et al., 2017). Since the late 1990s and early 2000s, several sensors have been launched to obtain data with a higher spatial resolution or

quality, such as SPOT-Vegetation and MODIS (Brown et al., 2006; Fensholt et al., 2006). The MODIS instruments onboard the Terra and Aqua satellites are considered as an extension and improvement of the AVHHR instruments, and they can provide an improved NDVI product from 2000, with superior characteristics (Didan et al., 2015; Huete et al., 1999). The MODIS NDVI product is suitable for vegetation monitoring due to the advanced spectral characteristics of the sensors, which were primarily designed for the study of vegetation and the land surface. Benefiting from the standardized MODIS

product production system established by NASA, the spectral bands used to calculate the MOD13 NDVI products have been well processed to reflect the true surface spectral characteristics (Huete et al., 2002). The advanced processing includes atmospheric correction, radiometric calibration, geometric distortion reduction, and the maximum value composite procedure (Didan et al., 2015). As a result, the MODIS NDVI product has a better data quality than the AVHRR and other



long-term NDVI products (Fensholt et al., 2009). The new MODIS Collection 6 data also features several improvements in
the retrieval algorithm, including the correction of the major sensor degradation impacts identified in previous studies (Wang
et al., 2012; Zhang et al., 2017b). The MOD13 products provide NDVI time-series data with a time interval of 16 days or
one month, at varied spatial resolutions of 250 m (primary resolution), 500 m, 1 km, and 0.05°. Given the superior spatio-
temporal resolution and better data quality, the MODIS NDVI products have been extensively applied in numerous
vegetation-related studies at regional and global scales (Guan et al., 2019; Kern et al., 2020). However, due to the fact that
MODIS only has nearly 20 years of observations less than AVHRR, it cannot meet the demands of three or four decades
long investigation of vegetation before the 21th century.

It can be seen from the above that there is an obvious trade-off in the current long-term NDVI products. AVHRR is the only
very long-term system with a time span from the early 1980s, but its coarse resolution usually leads to dilution of NDVI
signals in regional studies, and the data quality is relatively poor because of the lack of onboard calibration mechanisms and
orbital drift. Although other products, such as the widely applied MODIS data, can provide a higher spatial resolution,
historical observations before 2000 are unavailable and thus reducing the temporal coverage and the meaningfulness of
obtained analysis. As a result, it remains a challenge to produce a consistent long-term NDVI time series across the sequence
of multiple systems with their different sensor characteristics (Fensholt and Proud, 2012; Tian et al., 2015). Numerous
efforts have been made to construct continuous NDVI time series utilizing multi-sensor observations, such as the Vegetation
Index and Phenology Laboratory (VIP Lab) project (Barreto-Munoz, 2013; Brown et al., 2008; Liang et al., 2016). Limited
attempts have also been made to composite continuous long-term datasets based on the AVHRR GIMMS/GIMMS 3g
datasets and MODIS MOD13 products (Barreto-Munoz, 2013; Brown et al., 2008; Gao, 2000; Mao et al., 2012; Van
Leeuwen et al., 2006). However, the previous studies have mainly concentrated on eliminating the data gaps and improving
the quality of the AVHRR data, and the MODIS data have usually been resampled to a coarser resolution, to match the
former one. As a result, it is still difficult to explore the detailed spatial distribution of decades-long vegetation change, due
to the limited spatial resolution. Since the primary spatial resolution of AVHRR data is 1.1 km, it is reasonable to recover its
spatial pattern at the raw resolution to match the MODIS data, instead of resampling MODIS to a coarser resolution.
Enhancing the spatial resolution of remote sensing data can not only provide more abundant land surface contexts, but can
also improve the applicability of data in different scales by better matching the footprint of the ground observations (Chen,
1999; Reich et al., 1999; Xie and Li, 2020a). For example, it has been shown in many studies that the best footprint for flux
sites is approximately 1 km (Zhou et al., 2016). Previous studies also have proved that remote sensing data with a higher
spatial resolution can reduce the uncertainty and errors in estimating vegetation-related parameters, such as vegetation
productivity (Guan et al., 2017; Reich et al., 1999; Xie and Li, 2020b). Overall, it is of great importance to derive a long-
term global NDVI time series with the raw AVHRR spatial resolution and temporal coverage.

In this study, a 1-km long-term global NDVI product since 1982 is produced by fusing the MODIS and AVHRR products,
and the accuracy is comprehensively evaluated. The main aims of this study are: (1) to derive a MODIS-like NDVI time





series before 2000, with the spatial resolution and the data characteristics similar to MODIS; (2) to carefully assess the applicability of the derived NDVI time series with regard to the spatial distribution and temporal variation.

## 2 Data and pre-processing

The MODIS MOD13A2 NDVI product and the AVHRR GIMMS 3g NDVI product were selected as the basis to produce the 1-km NDVI products since 1982 (Huete et al., 2002; Pinzon and Tucker, 2014). LAC and HRPT AVHRR data were employed to verify the reliability of the new NDVI time series.

### 2.1 AVHRR GIMMS 3g NDVI product

The GIMMS 3g NDVI dataset is the most widely used global AVHRR NDVI dataset, created within the framework of the
GIMMS project, with advanced characteristics in time span, spatial continuity, and inter-calibration of different satellites (Tucker et al., 2005). It has been declared that the GIMMS NDVI is a non-stationary time series and must be recalculated every time if more recent years of data are added (Pinzon and Tucker, 2014; Tian et al., 2015). Compared to the earlier version of the GIMMS NDVI time series without data from the NOAA-17 and 18 satellites, the latest version of the GIMMS 3g data covers a more extended time period, from July 1981 to December 2014. The data have been carefully processed to
reduce the interferential effects, such as calibration loss, orbital drift, cloud cover, volcanic eruptions, etc. The discontinuity between the AVHRR/2 and AVHRR/3 sensors was accounted for using the Bayesian analysis method, with the help of SeaWiFS and SPOT-VGT NDVI data, and the detection and calibration of snowmelt were improved using the arctic growing season data rather than data from the entire year (Pinzon and Tucker, 2014). The spatial resolution of the GIMMS 3g product is 1/12 degree (nearly 8 km × 8 km), and the temporal resolution is half a month. Since the original data were
provided with the geographic projection, they were transformed into sinusoidal projections to match the MODIS data. What is more, the maximum value composite (MVC) technique was further employed to obtain the monthly GIMMS 3g NDVI.

### 2.2 MODIS MOD13A3 NDVI products

The MODIS instruments can observe the Earth surface twice a day at 10:30 am and 1:30 pm local crossing time, respectively, on the Terra and Aqua satellites, and with a repeat cycle of 16 days and a global coverage of approximately one day. Thus,
NASA provides various authoritative NDVI products with a temporal resolution of 16 days or monthly, which have been composited using the MVC method based on the daily data (Didan et al., 2015; Huete et al., 1999). This processing can obtain a more reasonable NDVI value to represent the vegetation status in the time period, because it is considered that cloud and other contamination usually leads to a lower NDVI value. The MOD13A3 collection is the monthly dataset with a spatial resolution of 1 km × 1 km. The data from 2000 to 2014 were all downloaded as the basis to composite the new NDVI
time series, while the data from January 2000 were not included because no data were acquired. The MODIS data are gridded using the standardized MODIS grid, which is a sinusoidal grid with 10 degrees by 10 degrees in both the latitudinal





and longitudinal directions at the equator. There are, in total, 460 non-fill tiles globally, and it was counted that, in total, there are 285 tiles for the land surface, including the large islands. Quality assurance (QA) data are also included in the dataset, and were applied to mark the contaminated data in the process of temporal smoothing of the NDVI data.

**2.3 Other data**

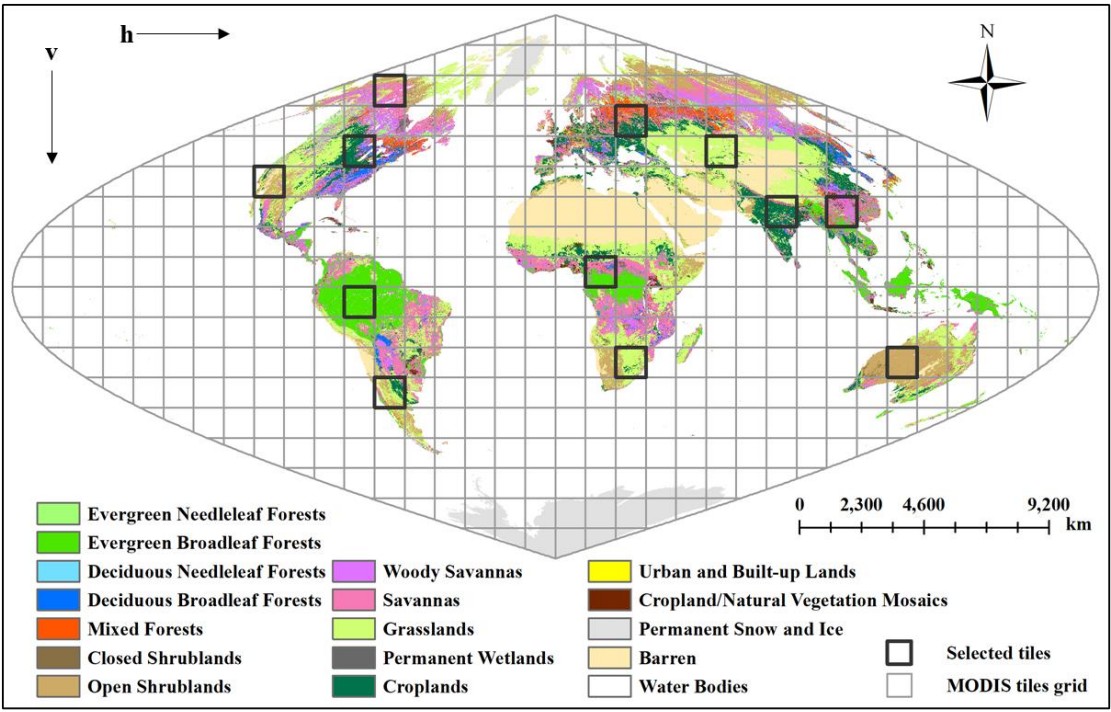

**Figure 1. Global MCD12Q1 land-cover map and the locations of the 12 selected tiles for the temporal stability validation of product performance with different vegetation major covers and latitude.**

Since this study aims to recover the GIMMS 3g NDVI to the raw resolution of AVHRR, the limited LAC and HRPT
AVHRR data with the raw resolution of 1.1 km are the ideal data to evaluate the fusion results. Since February 1992, agreements on HRPT were negotiated with various international organizations, which ensured the recording of the raw 1-km AVHRR data globally (Eidenshink, 2006; Eidenshink and Faundeen, 1994). Combining the observations at the HRPT ground station network, some global prototype 10-day composite products were produced by the USGS Earth Resources Observation Systems (EROS) Data Center (https://www.usgs.gov/centers/eros), in a project that was named the Global Land
1-KM AVHRR Project (GLAP). However, this project only worked for limited time periods, i.e., from 1 April 1992 to 31 October 1993, and from February 1995 to May 1996. As these data are not temporally continuous and there are some months missing in each year, we chose the data from February 1995 to January 1996, as a whole year, to assess the global spatial patterns of the fusion products. Furthermore, based on the LAC data received by stations in the conterminous U.S. and Alaska, USGS also produced weekly and bi-weekly 1-km AVHRR NDVI composites (ANCUS) in this region since the



NOAA 11 satellite launched in the year of 1989 (Eidenshink, 2006). In this study, the data from 1989 to 1993 were employed to validate the temporal change of the fused NDVI product, and in order to undertake a comparison, the data from 2001 to 2005 were used to assess the difference with the MODIS data.

The MCD12Q1 land-cover data were also applied to distinguish the performance of the new NDVI time series in different ecological communities (Sulla-Menashe et al., 2019). These data combine observations from sensors on both the Terra and

Aqua satellites, with a spatial resolution of 1 km, as shown in Fig. 1. The International Geosphere-Biosphere Program (IGBP) global vegetation classification scheme was selected in this study: ENF: evergreen needleleaf forest; EBF: evergreen broadleaf forest; DNF: deciduous needleleaf forest; DBF: deciduous broadleaf forest; MF: mixed forest; CSL: closed shrub land; OSH: open shrub land; WSA: woody savannas; SA: savannas; GL: grass land; PW: permanent wetlands; CRO: crop land. Images were selected to validate the accuracy of the NDVI products among 14 land-cover classes.

## 3 Method

### 3.1 Composition of the Long-term Global NDVI Series

Since the MODIS and AVHRR NDVI products have their respective superiorities in either the long time span or spatial resolution and data quality, a multi-step framework was designed to combine these respective advantages. Three processes were conducted to eliminate the problems in the data quality, sensor difference, and spatial resolution, respectively are the

temporal filtering, pixel-by-pixel normalization, and spatio-temporal fusion method.

### 3.1.1 Temporal filtering

The data quality of long-term NDVI products is usually impacted by unwanted noise and errors caused by cloud presence and other atmospheric contamination (Cao et al., 2018; Nagol et al., 2009; Yang et al., 2015). In order to obtain a high-quality NDVI time series, an improved trend filtering method was applied to denoise the GIMMS 3g and MODIS time series,

which combined the respective advantages of the $l_1$ trend filtering and Whittaker filtering method (Chu et al., 2021; Li et al., 2020). The Whittaker filtering method has been widely applied in the NDVI time-series data denoising, and $l_1$ trend filtering method substitutes $l_2$-norm used in the Whittaker filter method with $l_1$-norm (Kim et al., 2009; Montemerlo et al., 2002). The $l_2$-norm tends to obtain smooth results without noise impacts, and the $l_1$-norm can avoid over-smoothing in the key points but with some residual noise (Kim et al., 2009). As a result, the improved trend filtering combined advantages of the two norms

by using the $p$-norm in the regularization term, minimizing the loss of useful information while removing the noise. It has been proved to be a robust way to accurately identify and adjust the contaminated points within the time series (Li et al., 2020). This method can also achieve ideal time efficiency, which is very important for this study when conducting pixel-by-pixel processing of two global long-term datasets. Several pixels were sampled during the noise reduction, and it was found that the data quality of the MODIS dataset was clearly better than that of the GIMMS 3g time series, with much fewer

contaminated points.





### 3.1.2 Normalization

The NDVI between different sensors inevitably contains certain inconsistencies and gaps, due to the varied sensor-specific spectral band characteristics, types of surface features, atmospheric conditions, and other factors (Brown et al., 2008; Gan et al., 2014; Liang et al., 2016). Since the MODIS data have been carefully corrected for the atmospheric impacts, whereas there has been no atmospheric correction of the AVHRR data except for the two volcanic stratospheric aerosol periods, the MODIS data can reflect the true ground surface characteristics much better (Nagol et al., 2009; Pinzon and Tucker, 2014). In order to produce a continuous NDVI product utilizing the two datasets in different time periods, it is reasonable to normalize the AVHRR NDVI to the level of the MODIS data. Previous studies have concluded that the temporal trends derived from the GIMMS NDVI agree well with the MODIS data overall, and a pixel-by-pixel least-squares linear regression normalization model has been widely applied to narrow the discrepancy between the two datasets (Fensholt and Proud, 2012; Tian et al., 2015). For each pixel, the relationship between the two datasets could be obtained from the two vectors $y_M(x, y)$ and $x_A(x, y)$ comprised of the mutual pairwise time series from February 2000 to December 2014, as below:

$$y_M(x, y) = a(x, y) \cdot x_A(x, y) + b(x, y)$$

(1)

where $a(x, y)$ and $b(x, y)$ are the regression coefficients. The obtained relationship of each pixel was then applied to the corresponding pixel in the prolonged AVHRR data before 2000. Thus, the resulting AVHRR data series from 1982 to 1999 has MODIS-like NDVI values, with a coarser spatial resolution and similar radiometric characteristics. It has also been proved that the pixel-by-pixel normalization method is a more suitable method to normalize AVHRR and MODIS datasets, compared to the use of moving windows with different window sizes. This conclusion was based on the cross-validation conducted by randomly dividing the mutual pairwise images into ten folds, where nine folds were used to regress the relationship and the remaining one was applied to validate the predicted result.

### 3.1.3 Spatio-temporal fusion

After removing the sensor differences between the two data sources, differences still existed in the spatial resolution. Multi-sensor fusion is an efficient way to solve the problem and obtain data with the highest spatio-temporal resolution among all the data (He et al., 2018; Jiang et al., 2019; Shen et al., 2016). As such, it can be used to produce a data product with both the spatial resolution of MODIS data and the time span of AVHRR data, and is a popular way to obtain a continuous NDVI time series at the resolution of 30 m by fusing Landsat and MODIS data (He et al., 2018; Shen et al., 2016). Numerous studies have applied the spatial and temporal adaptive reflectance fusion model (STARFM) and the extended STARFM (ESTARFM) model to vegetation index prediction, and have proved its effectiveness in relieving the trade-off between different NDVI data sources (Gao et al., 2006; Zhu et al., 2010). The method used in this study was a spatio-temporal information fusion

235   method based on a non-local means filter, which has been proved to be able to achieve high accuracy and stability in different conditions (Cheng et al., 2017).

Multi-sensor fusion can allow us to predict NDVI data with the MODIS spatial resolution at an arbitrary time $t_k$, based on the AVHRR data at $t_k$ and the referenced MODIS and AVHRR images acquired at $t_0$. The prediction of fine-resolution NDVI before the year 2000 can be expressed as:

$$M\left(x_{w/2}, y_{w/2}, t_k\right) = \sum_{i=1}^{w} W_i \times \left(M\left(x_i, y_i, t_0\right) + A\left(x_i, y_i, t_k\right) - A\left(x_i, y_i, t_0\right)\right) \tag{2}$$

where $M$ and $A$ represent the fine-resolution MODIS data and coarse-resolution AVHRR data, respectively; $t_0$ is the acquisition date of the reference data; $t_k$ is the prediction date; $\left(x_{w/2}, y_{w/2}\right)$ is the location of the predicted pixel; $\left(x_i, y_i\right)$ denotes the pixel location; $w$ is the size of the moving window; and $W_i$ is the spatial weighting function. The innovation of the applied fusion algorithm is the more reasonable calculation of $W_i$, which takes full consideration of the spatial

245   relationship between pixels based on the concept of the non-local means filter. $W_i$ is calculated using:

$$W = \exp\{-\frac{G_a *(S+T)}{h}\} * 1/D \tag{3}$$

where $G_a$ is the Gaussian kernel in the non-local means filter; $h$ is the filter parameter; and $S$, $T$, and $D$ are the radiance difference, observed time difference, and geometric distance, respectively. The calculation of the three parameters can be achieved as follows:

$$S_i = \left|M\left(x_i, y_i, t_0\right) - A\left(x_i, y_i, t_0\right)\right| \tag{4}$$

$$T_i = \left|A\left(x_i, y_i, t_1\right) - A\left(x_i, y_i, t_0\right)\right| \tag{5}$$

$$D_i = 1 + \sqrt{\left(x_{w/2} - x_i\right)^2 + \left(y_{w/2} - y_i\right)^2} \Big/ \left(w/2\right) \tag{6}$$

where the variables are the same as in Eq. (2).

Since the referenced pairwise MODIS and AVHRR data are important for the fusion processes, a fusion strategy considering

255   the seasonal variation of vegetation was applied to obtain a better result. For the NDVI in each month, the data before 2000 were fused, referring to the pairwise data in the corresponding month in the year 2000 or 2001. Due to the fact that no data or the low data quality in January and February 2000, the reference data in these two months were selected from 2001, and the reference data in the other ten months were chosen from 2000.



## 3.2 Evaluation of NDVI datasets

### 3.2.1 Simulation experiment

Since this study aims to produce a new NDVI product, it is naturally essential to verify the applicability of the constructed dataset. Due to the fact that there are no actual MODIS data before the year 2000, it is difficult to examine the accuracy of the STFLNDVI results directly. As a result, a simulated experiment was first carried out to validate the feasibility of the fusion framework, based on the overlapping time series of MODIS and AVHRR data from 2000 to 2014. The NDVI data from 2000 to 2013 were simulated using exactly the same processes as described in Section 3.1, and the pairwise MODIS and AVHRR data from 2014 were taken as the reference data in the spatio-temporal fusion step. In this condition, the results could thus be assessed by comparing them with the true MODIS data during this period, using the correlation coefficient ($r$), mean absolute difference (MAD), and bias. Due to the fact that the fusion framework is expected to obtain MODIS-like data, so a better result is the fused data nearer to the true data. The accuracy of both the global spatial patterns and temporal variations was carefully assessed, and 12 MODIS tiles with different major vegetation cover were selected (as shown in Fig. 1) to prove the stability of the evaluation accuracy in different years and seasons.

### 3.2.2 Comparisons with true AVHRR data

Although the implemented simulated experiments can prove the feasibility of the fusion process framework, it is also necessary to directly assess the accuracy of the obtained product from 1982 to 1999. Since this study tends to construct a 1-km NDVI time series on the basis of the AVHRR GIMMS 3g product, the true AVHRR data with the raw resolution of 1.1-km would be the most suitable data for validation. Even though most of the AVHRR data are stored at a reduced resolution because of the huge data volume and limited storage capacity during the 1980–1990 period, attempts have been made to obtain some LAC and HRPT observations at the raw 1.1-km resolution for limited times and specific regions. These data provide the possibility to conduct the direct evaluation of the STFLNDVI product before 2000, including the spatial context and temporal variation comparison. Due to the applied normalization processes and the differences in processing, there would exist certain gaps between the radiance characteristic of fusion data and true AVHRR observations. As a result, an adjusted MAD (AMAD) is used to assess the data radiances differences between the two data, which is calculated as the absolute value of the difference between the MAD and bias by eliminating the systematic bias. The GLAP data from February 1996 to January 1996 were used to assess the global spatial consistency, and the ANCUS data from 1989–1993 and 2001–2005 were employed to evaluate the temporal consistency across the conterminous U.S. area.



## 4 Results

### 4.1 Reconstructed 1-km continuous global NDVI time series from 1982

Based on the multi-step processing, the spatio-temporal fusion-based long-term NDVI product (STFLNDVI) was reconstructed by improving the spatial resolution of the AVHRR NDVI time series to 1 km. This dataset combines the
respective advantages of MODIS and AVHRR data, with the long time coverage of AVHRR data and the spatial resolution of MODIS data. The values and radiometric characteristics are also carefully calibrated to be close to the MODIS product. As a result, this dataset can be regarded as an acceptable backward extension of MODIS data with consistent data characteristics. In the future, if the follow-up observation data are consistent with MODIS, they could be directly added to this product to form a spatiotemporal continuous NDVI dataset covering the time period from 1982 to date, which will be
helpful for global or regional vegetation studies. This dataset is also provided using the standardized MODIS sinusoidal grid with the same geographic projection to be consistent with the MODIS product. After lots of experiments, it was confirmed that two adjacent tiles of STFLNDVI data could be seamlessly mosaiced globally as the MODIS data. This 1-km long-term NDVI product has been publicly released and can be freely downloaded from https://zenodo.org/record/4734593.

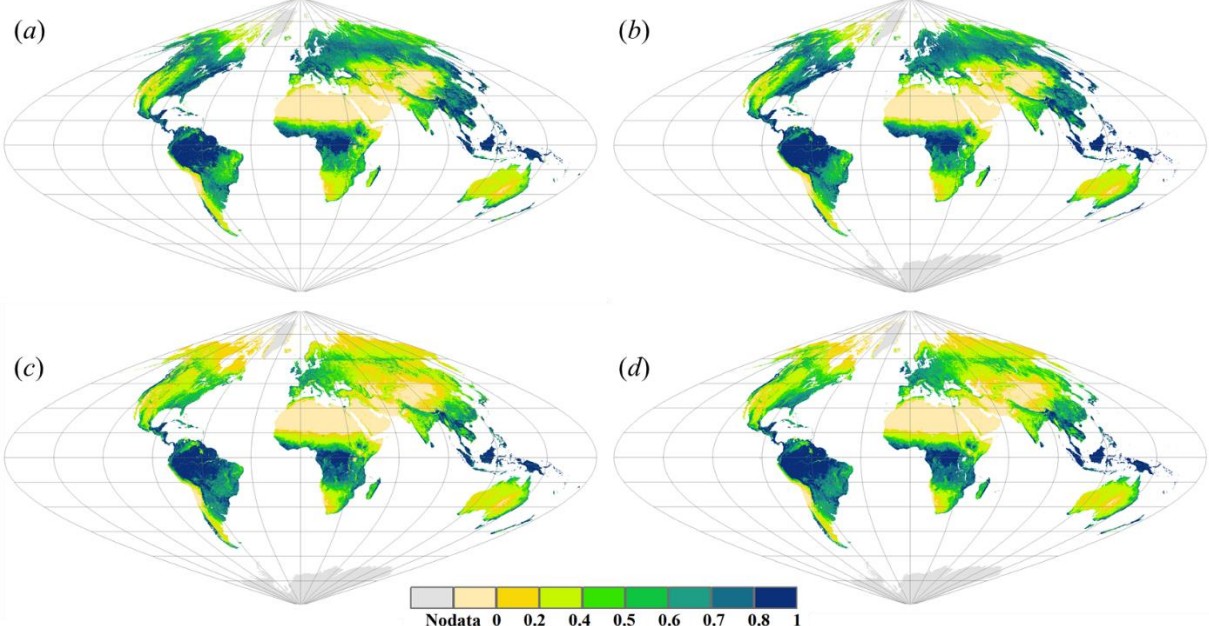

**Figure 2. Spatial distribution of the mean NDVI in the different seasons of 1990: (a) and (b) are the mean STFLNDVI and GIMMS 3g NDVI in the growing season for the Northern Hemisphere, respectively; (c) and (d) are the data in the off-season for the Northern Hemisphere. The growing season is from May to October, and the offseason is the other six months.**

The new STFLNDVI product has a similar global spatial distribution to the original AVHRR data, but with a higher spatial resolution and richer spatial details. In Fig. 2, the seasonal NDVI in the original AVHRR data and the fused STFLNDVI
product are compared. The year 1990 was divided into two seasons according to the vegetation status and climate in the





Northern Hemisphere, i.e., a growing season from May to October and an off-season in the remaining six months. It can be observed that the STFLNDVI product shows a similar overall spatial distribution to the original GIMMS 3g data, in both the growing season and the off-season. This indicates that the fusion procedure and other processes have not changed the global spatial pattern of the NDVI distribution, and the final product has the capacity to represent the true vegetation distribution.

Furthermore, the STFLNDVI data can provide significantly more spatial details than the original NDVI at the regional scale, which are similar to the true spatial patterns of the AVHRR and MODIS data, as compared in Sections 4.2.2 and 4.3.1.

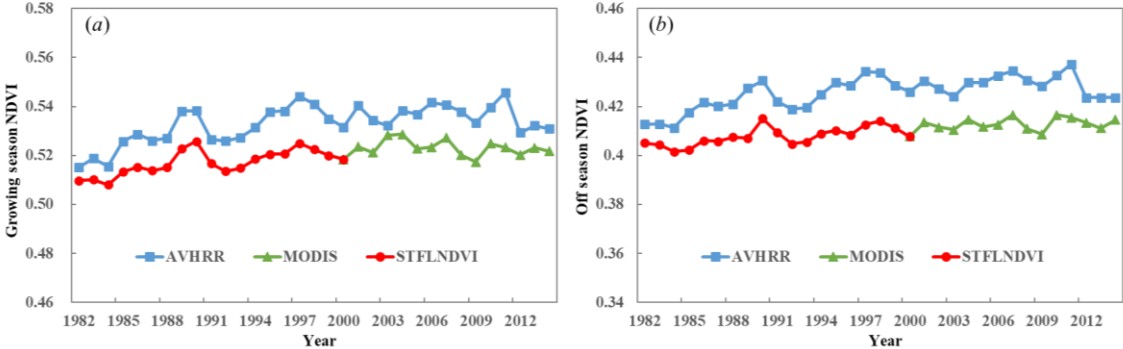

**Figure 3. Inter-annual variation of the global mean NDVI for the AVHRR GIMMS 3g, MODIS MOD13A3, and STFLNDVI data: (a) and (b) are the results in the growing season and off-season for the Northern Hemisphere, respectively.**

The inter-annual variations of the global mean NDVI for the AVHRR GIMMS 3g, MODIS MOD13A3, and STFLNDVI data are compared in one plot in Fig. 3. A significant gap can be observed between the AVHRR and MODIS data, in both the growing and off-season, which can be attributed to the lack of atmospheric correction for the AVHRR data, as well as the differences in the sensor characteristics, transit time, and other factors. The values of the AVHRR NDVI are generally higher than those of MODIS, with a mean difference of 0.0098 for the growing season and 0.013 for the off-season. Due to the

normalization processing adopted to make the data the same as MODIS, the difference is inherited by STFLNDVI, with generally lower values than the AVHRR data from 1982 to 1999. However, the inter-annual variations between the AVHRR and STFLNDVI data are very similar, which indicates that the fusion processing does not change the temporal variation of the NDVI. As a result, the data characteristics of STFLNDVI are identical to those of the MODIS data, and the temporal variation is inherited from the original AVHRR data, because it is the only record and resource to trust.





## 4.2 Simulated evaluation based on the overlapping time series

### 4.2.1 Evaluation of the temporal consistency

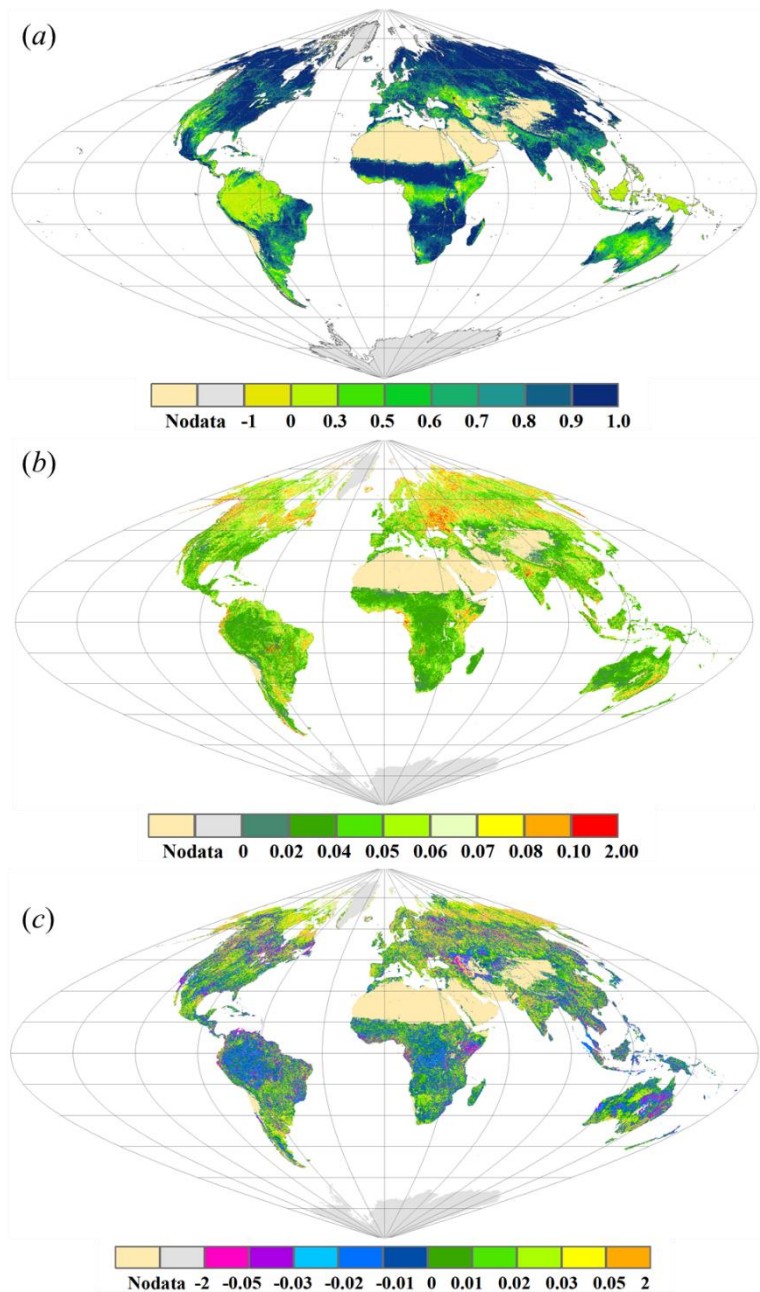

**Figure 4. Spatial distribution of the pixel-by-pixel temporal consistency between the fusion results and the true MODIS data during the overlapping period from 2000 to 2014: (a) *r*, (b) MAD, (c) bias.**





The temporal consistency of the fusion results in the simulated experiments was first evaluated pixel by pixel. The correlation coefficient ($r$), mean absolute difference (MAD), and bias of the temporal variation between the fusion result and the true MODIS data were calculated in each pixel, as shown in Fig. 4. It can be observed that the fusion results show similar variation characteristics to the true MODIS data in most areas, with a high $r$ and low MAD and bias. However, the performance of the fusion result is different in the Northern Hemisphere and Southern Hemisphere. In the Northern

Hemisphere, almost all areas show $r$ values of higher than 0.9, except for some small regions in western America and Central Asia with rare vegetation cover and a relatively low NDVI. Although high $r$ values can also be observed in most areas with high vegetation cover in the Southern Hemisphere, a worse relationship can be observed in Australia. This is induced by the low vegetation cover in this region. However, the global distribution of MAD and bias is almost the opposite. For the pixels in the Northern Hemisphere, especially in the high-latitude areas, the difference between the fusion result and

the true MODIS data is much higher than in the Southern Hemisphere. Large areas in the high-latitude Northern Hemisphere show a MAD value ranging from 0.06 to 0.10 and a bias ranging from ±0.03 to ±0.05, whereas the values in the Southern Hemisphere are mostly within 0.0–0.4 and ±0.02, respectively. The results are unsatisfactory in the area near the equator, where the famous and vital Amazon rain forest and Congo basin are located, with $r$ values generally less than 0.03. However, the two difference metrics are very low, with the MAD generally being less than 0.04 and the bias within ±0.02. This

indicates that although the temporal variations between the two data sources are not the same, the values are quite close. The spatial difference of the temporal consistency is mostly due to the original relationship between the AVHRR and MODIS data, which is discussed in Section 5.

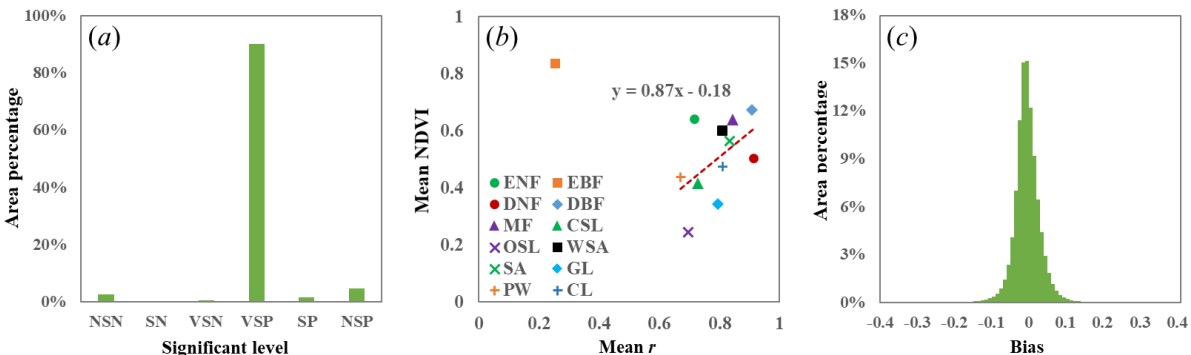

**Figure 5. Statistics of the pixel-by-pixel $r$ and bias: (a) the percentage of area for $r$ with different significance levels; (b) the mean $r$**
**and the mean NDVI of the different vegetation types; (c) statistical histogram of the global bias with a step size of 0.01. NSN: not significant negative ($r<0$, $p>0.05$); SN: significant negative ($r<0$, $0.01<p<0.05$); VSN: very significant negative ($r<0$, $p<0.01$); VSP: very significant positive ($r>0$, $p<0.01$); SP: significant positive ($r>0$, $0.01<p<0.05$); NSP: not significant positive ($r>0$, $p>0.05$).**

The statistics of the global $r$ and bias are demonstrated in Fig. 5. We divided the $r$ values into six categories according to the
pixel $r$ and $p$ values: very significant negative (VSN, $r<0$, $p<0.01$), significant negative (SN, $r<0$, $0.01<p<0.05$), not
significant negative (NSN, $r<0$, $p>0.05$), very significant positive (VSP, $r>0$, $p<0.01$), significant positive (SP, $r>0$,
$0.01<p<0.05$), and not significant positive (NSP, $r>0$, $p>0.05$). It can be found that about 92.41% of the global pixels show a





significant positive relationship between the fusion result and the true MODIS data, and 90.90% have a *p* value of less than

0.01. Only limited pixels showed a non-significant positive relationship or a negative relationship, most of which are

concentrated in the area around the equator. The *r* values for 12 types of vegetation were also calculated based on the

MCD12Q1 dataset for the year 2001. It can be observed that almost all the vegetation types have a high mean value of *r*,

except for the evergreen broad-leaved forest (EBF). This is because most of the EBF category is distributed in the rain forest

near the equator, and their seasonal amplitude is low compared to atmospheric-induced noise in the time series. If the EBF

points are removed, an obvious positive relationship between the mean NDVI value and the mean *r* can be observed among

the 11 vegetation types (*r*=0.52, *p*<0.05). This indicates that the vegetation with a higher NDVI usually shows a better

performance in the fusion results. The bias of the fusion data and true MODIS data is also very low, with 98.33% of the

global area showing a bias within the range of −0.1 to 0.1, and 91.80% in the range of −0.05 to 0.05. Even in the area near

the equator with low *r*, the bias between the fusion results and the true MODIS data is mostly less than 0.05. This reveals

that even though the temporal consistency is not very strong, the absolute differences in predicted NDVI values are quite

small.

**4.2.2 Evaluation of the spatial distribution**

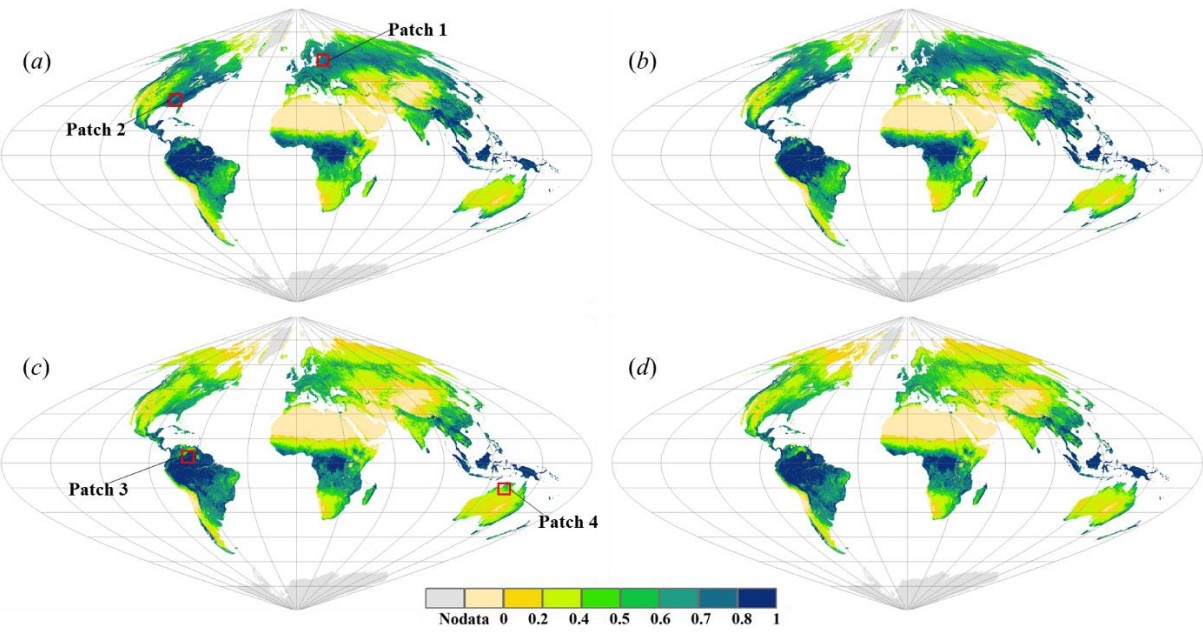

**Figure 6. Spatial comparisons of the seasonal NDVI of the fusion results and the true MODIS product in the different seasons of 2002: (a) and (b) are the mean values of the fusion data and true MODIS NDVI in the growing season, respectively; and (c) and (d) are the data in the off-season.**

Since the true MODIS data were available in the simulated experiments, the spatial distribution of the fusion results could be

evaluated. The results for 2002 are selected for display in Fig. 6, because this year is far from the reference year of 2014 and

has a full year of data. Similar to the spatial comparison between the STFLNDVI product and the AVHRR data, the mean NDVI values in the growing and off-seasons are compared globally. It can be observed that, in both seasons, the overall spatial distribution of the NDVI in the fusion results is very similar to that of the true MODIS data, and it is difficult to find any difference between the two figures. We calculated the $r$ and bias values for the two sets of data. The $r$ between the fusion result and the true MODIS data is almost the same for the two seasons, respectively, at 0.87 and 0.88. The bias is also very low, at −0.0058 and 0.0097 for the growing season and off-season, respectively. These results reveal that the fusion process can effectively improve the spatial resolution of the NDVI product, and the results show similar spatial patterns as the true MODIS data.

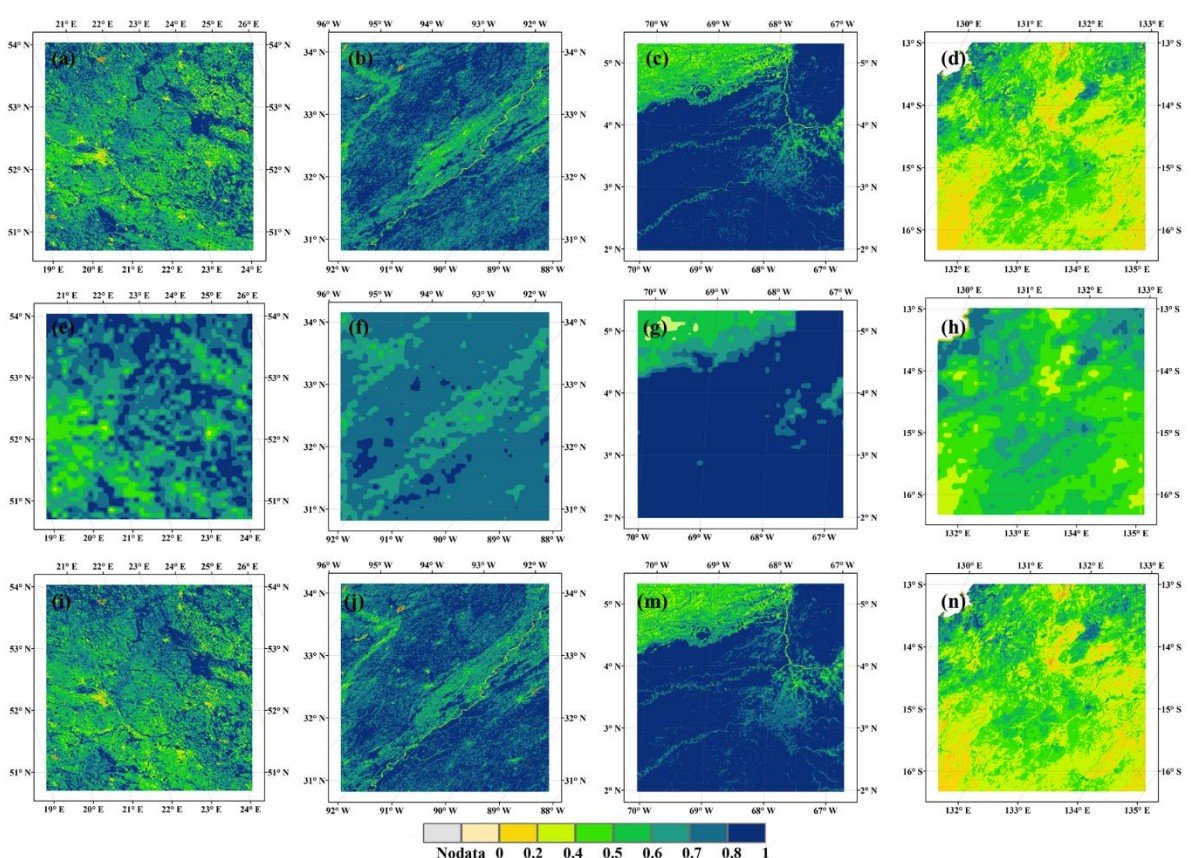

**Figure 7. Subset spatial comparisons of the different data. From left to right are Patches 1, 2, 3, and 4 in Fig. 6, respectively; and (a)-(d) are the true MODIS data, (e)-(h) are the original AVHRR data, and (i)-(n) are the fusion results.**

Furthermore, we also selected four typical areas for a close comparison in Fig. 7, with different vegetation types and different heterogeneity of land cover. Patches 1, 2, 3, and 4 were selected in different years and different months, i.e., May 2001, August 2000, February 2003, and November 2002. These four years are all far from the reference year, and are randomly distributed in different months. The results indicate that the fusion data and the MODIS data can clearly provide



Earth System
Science
Data

much more spatial information, and the fusion results show very similar spatial details to the true MODIS data. Although some subtle differences can be found in the edges of some features, these are unlikely to affect the analysis of the vegetation distribution. Even for Patch 3 located in the Amazon area, although the temporal consistency is relatively poor, the spatial

distribution is almost the same. These results indicate that the similarity of the two data sources is very close in different times and areas, and also in different areas with varied vegetation cover and spatial heterogeneity.

### 4.2.3 Temporal stability of product performance

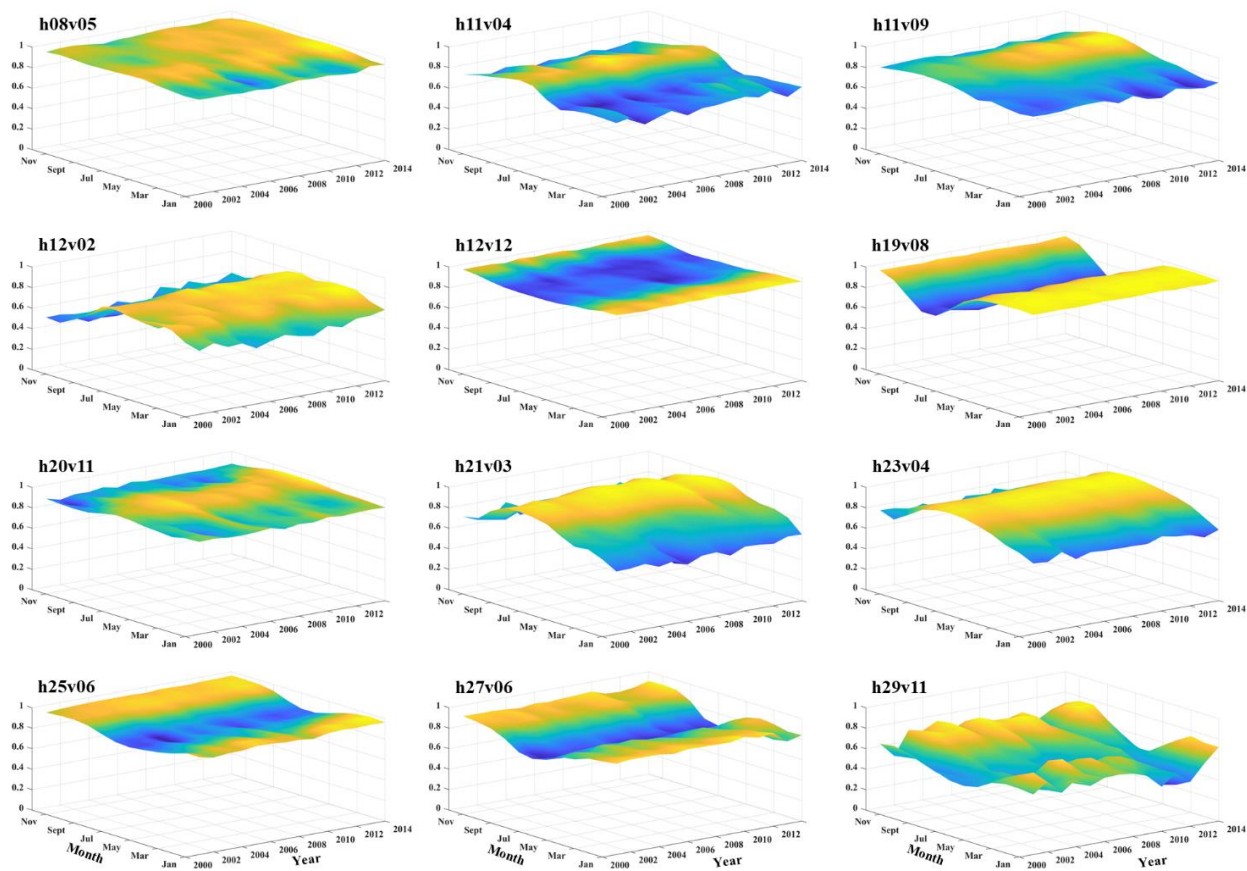

**Figure 8. Intra-annual and seasonal variation of the *r* between the fusion data and true MODIS NDVI in the 12 selected tiles. The**
**X-axis is the year, the Y-axis is the month, and the Z-axis is the value of *r*.**

Due to the STFLNDVI product in each month from 1982 to 1999 being fused based on the pairwise reference data from the year 2000 or 2001, it is important to evaluate the stability of the accuracy over the years and months. It is also important to assess whether the fusion data can obtain a stable accuracy for a year that is far from the reference data. Based on the results of the simulated experiments, it was possible to reveal this issue by comparing the fusion data with the true MODIS data in

different years and different months. We, therefore, selected 12 MODIS tiles of data (as shown in Fig. 1), randomly





distributed around the world and dominated by different vegetation types, and the $r$ and MAD in each time phase were calculated to represent the accuracy of the fusion results. The intra-annual and seasonal variation of the $r$ and MAD values is demonstrated in Fig. 8 and Fig. 9, respectively. The transects along the X-axis are the seasonal variations in different years, and the transects along the Y-axis are the intra-annual variations in different months. It can be observed that the fusion

results show good spatial consistency with the true MODIS data in almost all 12 tiles, with $r$ values of generally more than 0.8 and MAD values of generally less than 0.08. In most cases, the $r$ is greater than 0.9, and the MAD is less than 0.04. The $r$ is lower in the area of tile h29v11, where are mainly distributed by sparse shrubland with a relatively low mean annual NDVI of 0.23. However, even in this condition, the $r$ is also generally higher than 0.65, and the MAD is generally around 0.03. The MAD is higher in some months in the highly vegetated regions with high latitude, where the seasonal variation of

the NDVI is much more obvious. The value of the MAD is still around 0.08 in these cases. There are limited differences in the seasonal variation of these two indices in many of the tiles distributed in low latitudes or areas with stable monthly NDVI. Overall, the fusion results are generally good in almost all the areas, with high $r$ and low MAD values.

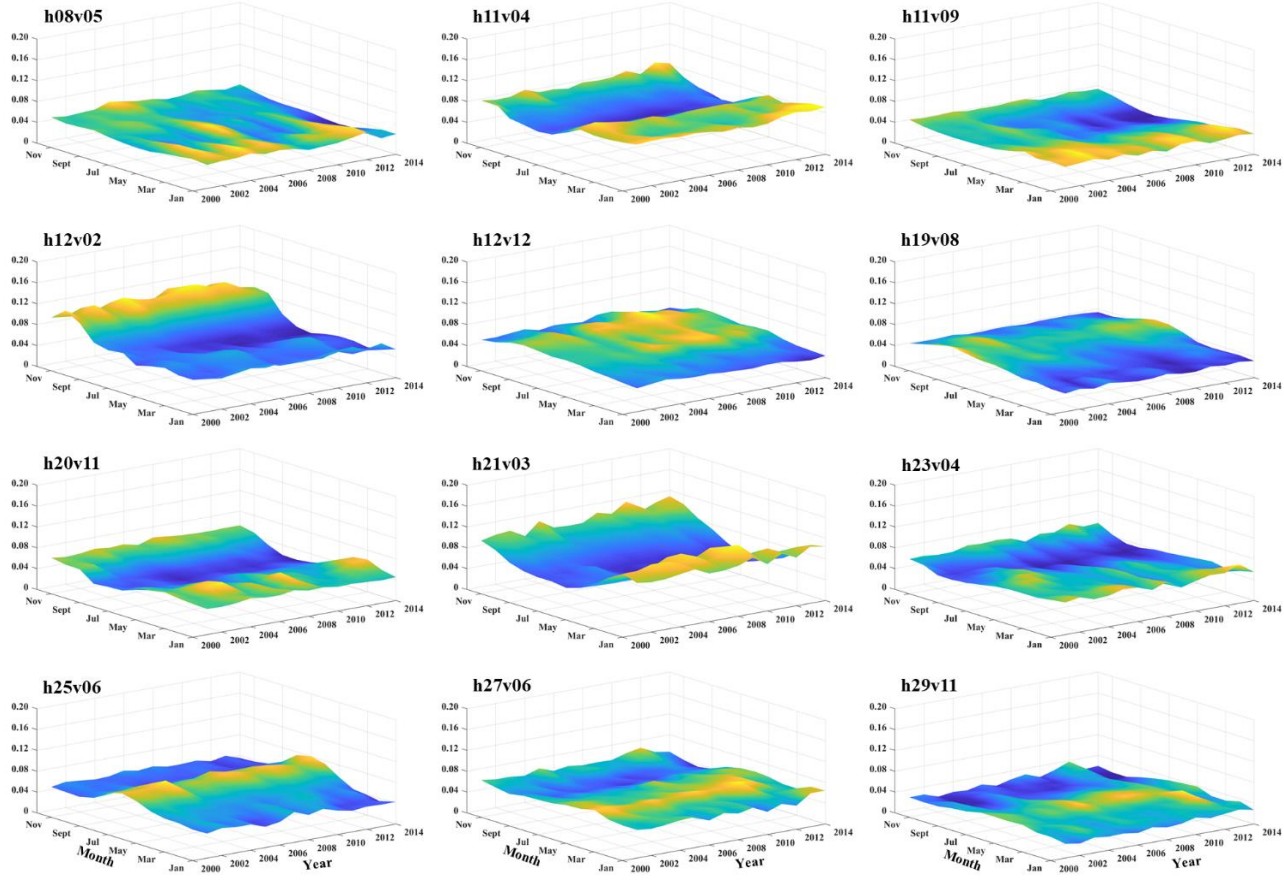

**Figure 9. The intra-annual and seasonal variation of the MAD between the fusion data and true MODIS data in the 12 selected**
**tiles. The X-axis is the year, the Y-axis is the month, and the Z-axis is the value of the MAD.**





What is more, both the *r* and the MAD show stable intra-annual variations with low fluctuation in almost all the tiles. The fluctuation of *r* is relatively high in tile h29v11, but it does not decrease from 2014 to 2000. Furthermore, the variation of the MAD is also very stable in this tile. There are no annual variation trends in the changes of *r* and MAD in all regions and months. This indicates that the processes used in this study can obtain a stable fusion accuracy when the fusion year is far

from the reference data, in different areas and months. Therefore, the STFLNDVI product for 1982 is expected to be as reliable as it is for 1999, and the temporal interval between the fusion data and the reference data is expected to have little impact on the fusion accuracy. This could mostly be due to the coarse spatial resolution of the MODIS and AVHRR data, in that the spatial texture is not as abundant as in Landsat data or data with an even higher spatial resolution. The fusion method applied in this study is processed as a non-local treatment, which can naturally obtain stable results in the condition of the

spatial difference between the two data sources being not very large.

From the above analysis, we can conclude that the proposed fusion framework can obtain accurate and stable NDVI data that very similar to MODIS data, in both the spatial distribution and temporal variation.

### 4.3 Evaluation against true AVHRR data

#### 4.3.1 Comparison of the spatial patterns

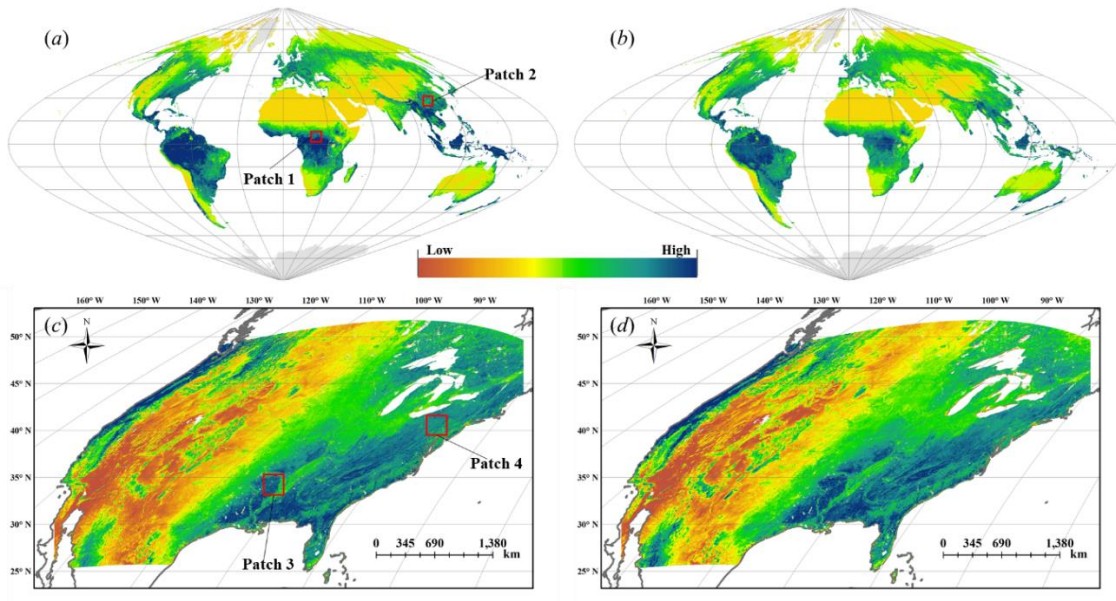


**Figure 10. Comparison of the spatial patterns: (a) and (b) are the mean NDVI from February 1995 to January 1996 for the STFLNDVI product and the true GLAP observations, respectively; and (c) and (d) are, respectively, the STFLNDVI product and the ANCUS observations for 1992.**

Except for the radiance characteristics, which are similar to MODIS, the spatial patterns and temporal variations of the

STFLNDVI product should be the same as the true AVHRR observations. Due to the normalization processes conducted to



convert STFLNDVI to be similar to the MODIS data, certain differences exist in the NDVI values when compared to the true AVHRR observations. These differences are difficult to eliminate because they are synthetically caused by the sensor characteristics, original data generation, and the processes conducted in the STFLNDVI framework. As a result, it is not appropriate to compare the spatial distribution using the raw NDVI values, and a comparison with stretched color is provided

here, which can directly reflect the NDVI distribution in the different data. In Fig. 10 (a) and (b), the mean NDVI from February 1995 to January 1996 as a whole year is compared between STFLNDVI and the true GLAP observations, in order to assess the global distribution. The permanent glacier and desert areas are masked with light yellow. It can be observed that there is great consistency in the global spatial patterns of the two data types. Another true AVHRR observation product (ANCUS) for the region of the conterminous U.S. was also applied to assess the spatial consistency of the fusion data, as

shown in Fig. 10 (c) and (d). Almost the same NDVI distribution can be found for the two data types, which indicates good overall performances in the spatial distribution of STFLNDVI.

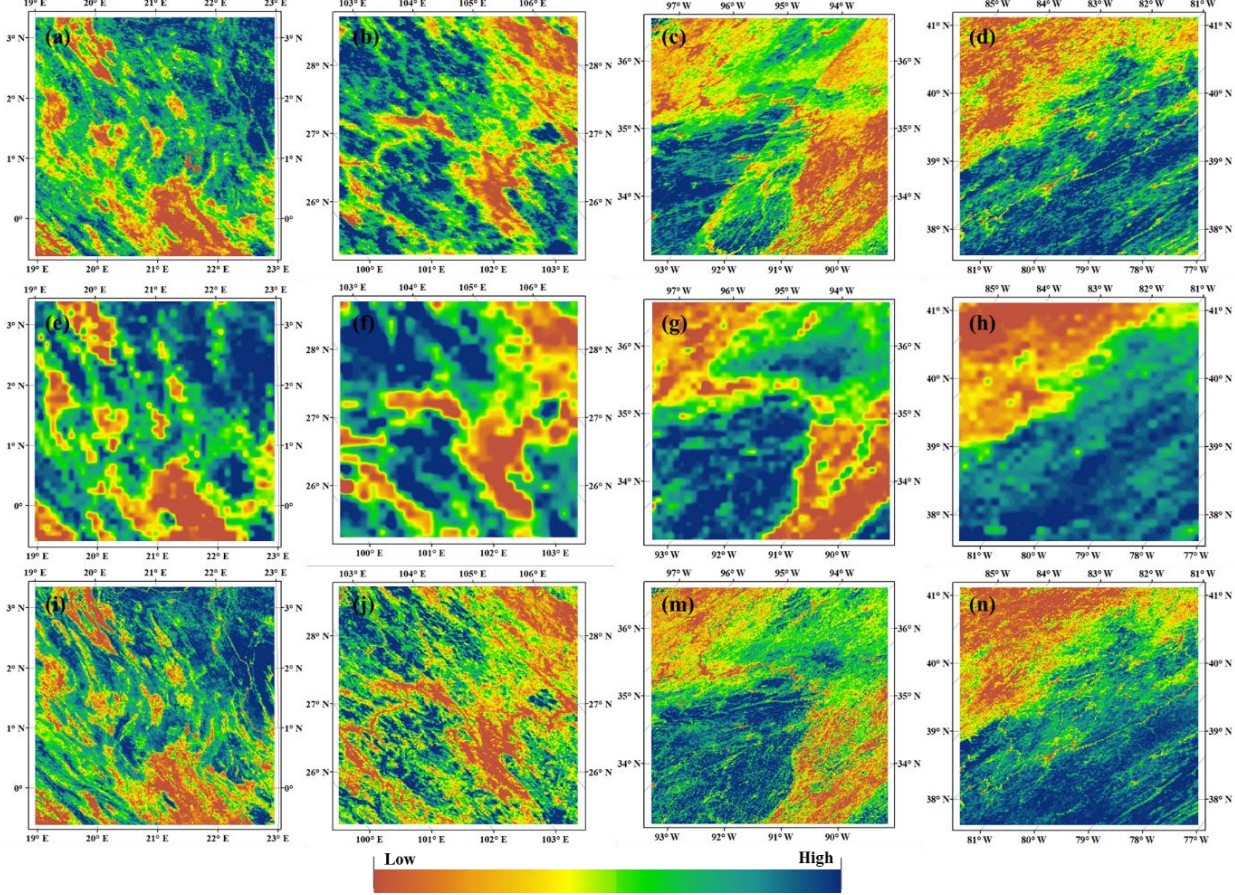

**Figure 11. Zoomed spatial comparisons of the different data: from left to right are the enlarged images of Patches 1, 2, 3, and 4 from Fig. 10, respectively; and (a)-(d) are the true GLAP/ANCUS AVHRR observations, (e)-(h) are the original GIMMS 3g data,**
**and (i)-(n) are the STFLNDVI results.**



In order to further validate the spatial patterns in the fusion dataset at the regional scale, which is the main improvement of the fusion data compared to the original GIMMS 3g product, four patches were selected with the size of $400 \times 400$ pixels. The locations of these patches are marked in Fig. 10 (a) and (c), and the comparison results are shown in Fig. 11. The first two patches are the comparison with the GLAP data from April and September 1995, and the last two patches are the

comparison with the ANCUS data from January 1992 and July 1990. All the results indicate that the spatial patterns in the STFLNDVI results are very similar to the true observations, and almost all the details can be found. Although the overall spatial distribution of the original GIMMS 3g data is similar, almost all the small features are lost compared to the other two data types. This proves that the STFLNDVI data can enhance the spatial resolution and provide more detailed patterns, which will be very helpful in regional vegetation studies. Furthermore, in most cases, the true AVHRR observations in the

GLAP/ANCUS data do not appear as clear as the fused STFLNDVI data, even though they are at the same spatial resolution. The comparison of Patch 3 in Fig. 11 also indicates the problem of obvious gaps in the ANCUS data, as a result of the poor correction between the different observations. Evident gaps will therefore appear when it is necessary to mosaic observations from different instruments or different times.

### 4.3.2 Evaluation of the global spatial consistency

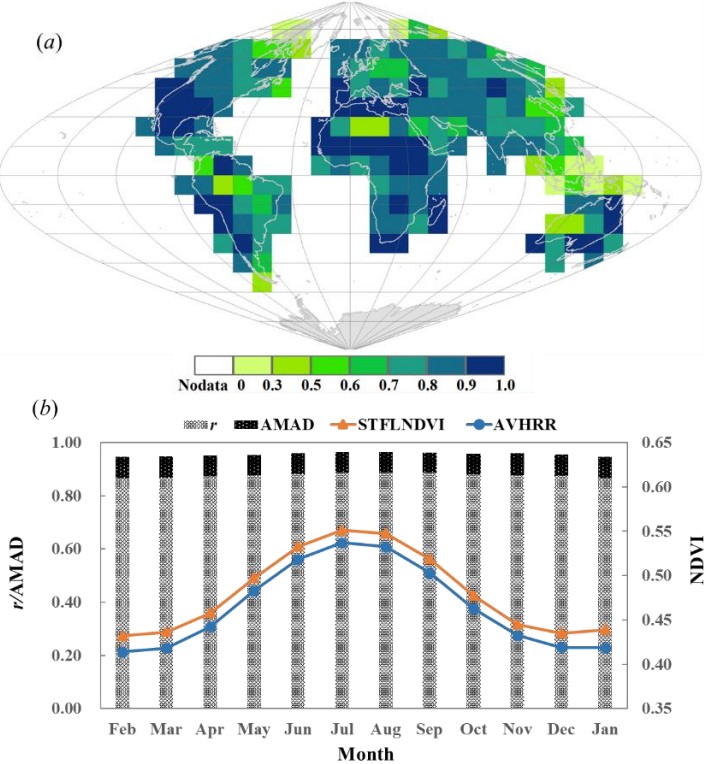


**Figure 12. (a) Comparison of the *r* between the STFLNDVI product and the true GLAP data in each MODIS tile; (b) intra-annual variation of the mean NDVI values and the relationship between the two data types in each month.**



Due to the limited observations of the true GLAP data, only a global quantitative analysis of the spatial consistency and its seasonal variation could be conducted. In order to further evaluate the global spatial consistency of the STFLNDVI product,

the global mean NDVI data from February 1995 to January 1996 were divided into different patches according to the MODIS tiles. Considering the evident radiometric differences between the STFLNDVI product and the true GLAP observations, the correlation coefficient $r$ is a more suitable index to evaluate the consistency between them, because it is not related to the magnitude of the data. Therefore, the $r$ values between the two data types in all the patches with mostly land area coverage were calculated, and the results are shown in Fig. 12 (a). It can be observed that the two datasets show good

consistency in almost all the MODIS tiles, with the $r$ being mostly higher than 0.7. More than 93.2% of the tiles show a significant positive correlation ($r>0$, $P<0.05$), and 85.92% show a very significant positive correlation ($r<0$, $P<0.01$). Only a few tiles show a low correlation between the two data types, and are mostly distributed in the island areas around the equator. These regions always have frequent rainy days and fragmented small land areas, so that it is harder to capture the detailed spatial patterns.

The intra-annual variation of the global spatial consistency between the fusion data and true GLAP observations was also evaluated. Fig. 12 (b) shows the correlation coefficients and the AMAD values between these two data types in each month from February 1995 to January 1996, as well as the mean NDVI values. Due to there was an obviously stable system bias around 0.03 in each month for the GLAP and STFLNDVI data, it is necessary to remove the bias and the AMAD calculated by MAD minus bias is used to evaluate the differences. The results indicated that the seasonal variation of the mean values

for STFLNDVI and the true observations are almost the same, with the $r$ between them being higher than 0.99 ($P<0.01$). Besides, a stable intra-annual variation of the two indices can also be observed, with a high $r$ of around 0.88 and a low AMAD of around 0.078. Therefore, the fusion data have good spatial consistency with the true observations, and it can effectively capture the seasonal variations of the vegetation in each month at a global scale.

**4.3.3 Evaluation of the temporal consistency across the conterminous U.S.**

Since the limited GLAP observations cannot be used to form a multi-year time series, five years of ANCUS observations across the conterminous U.S. from 1989 to 1993 were selected to assess the inter-annual variation of the fusion NDVI. In order to evaluate the similarity between the MODIS and fusion products, ANCUS data with the same temporal length from 2000 to 2004 were also employed to calculate the correlation with the MODIS observations, which was compared with the correlation of STFLNDVI. The spatial distribution of the $r$ of each pixel from 1989 to 1993 for STFLNDVI and the true

AVHRR observations is illustrated in Fig. 13 (a), and Fig. 13 (b) is for the results for the MODIS data from 2001 to 2005. Good consistency between the original MODIS and AVHRR data can be observed, and this consistency has been favorably inherited by STFLNDVI. Most of the pixels show very high correlations, with values greater than 0.8 for both the STFLNDVI and MODIS data. About 93.42% of the MODIS pixels show a significant positive correlation ($r>0$, $P<0.05$), and 88.95% show a very significant positive correlation ($r>0$, $P<0.01$), while the ratios are 90.80% and 85.32% for the

STFLNDVI product. Only a few pixels show non-significant low correlation, which are mostly distributed in the western



and southern regions with a relatively low vegetation coverage. Simultaneously, the pixels showing a reduction of significant correlation in STFLNDVI compared to the MODIS data are also mostly distributed in these regions. The complex topography may explain the poor performance in these regions (Hao et al., 2018; Matsushita et al., 2007).

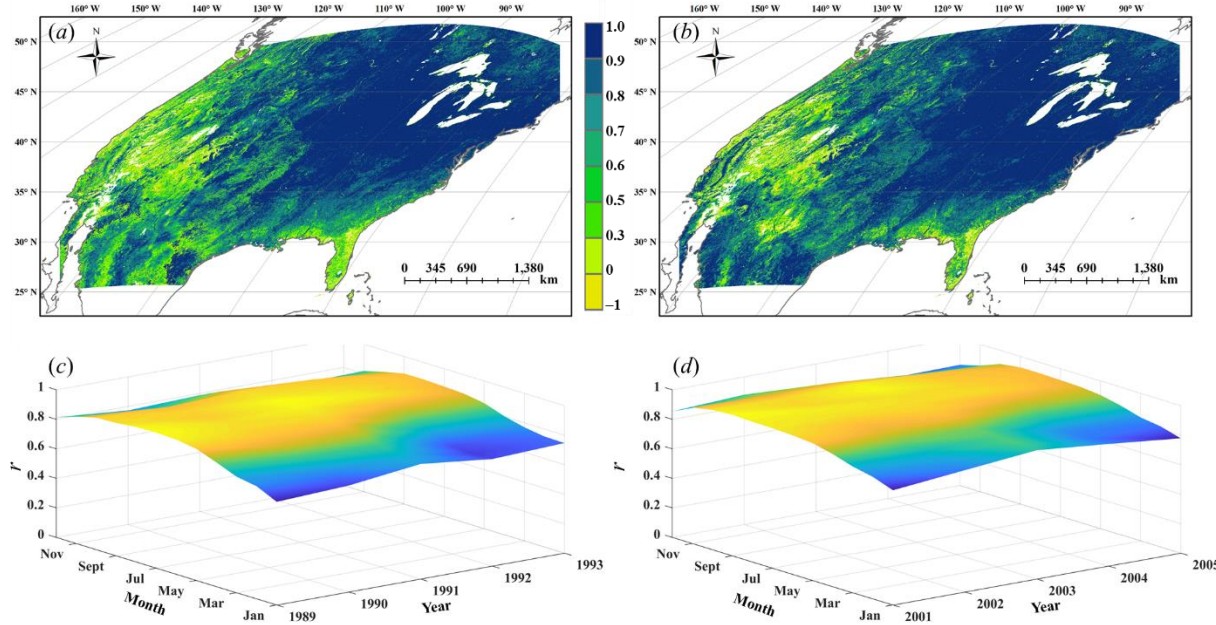

**Figure 13. Temporal consistency of the STFLNDVI and MODIS data compared to the true ANCUS NDVI: (a) and (b) are the spatial distributions of the *r* for the STFLNDVI product from 1989 to 1993 and the MODIS data from 2001 to 2005, respectively; and (c) and (d) are the variation of *r* in different years and months.**

Fig. 13 (c) and (d) also demonstrate the inter-annual and intra-annual variations of the spatial distribution consistency between STFLNDVI/MODIS and the AVHRR observations. Both the STFLNDVI and MODIS data show a high correlation
with the AVHRR data in different years and different months, with a mean value of 0.84 for the fusion data and 0.88 for the MODIS data. Furthermore, the variations across different years and different months are almost the same for the STFLNDVI and MODIS data, with stable inter-annual variations and slight seasonal changes. The *r* value is generally higher in the summer months and lower in the winter months, for both the STFLNDVI and MODIS data. As a result, it can be concluded that STFLNDVI agrees well with the true AVHRR observations, in both the spatial distribution and temporal variation, and
its characteristics are almost the same as the MODIS data. This indicates that the fusion data represent an effective extension of the MODIS data before the year 2000, with similar data characteristics, and the fusion data are comparable with the true observations during this period.



# 5 Discussion

## 5.1 Reasons for the heterogeneity of the spatio-temporal consistency

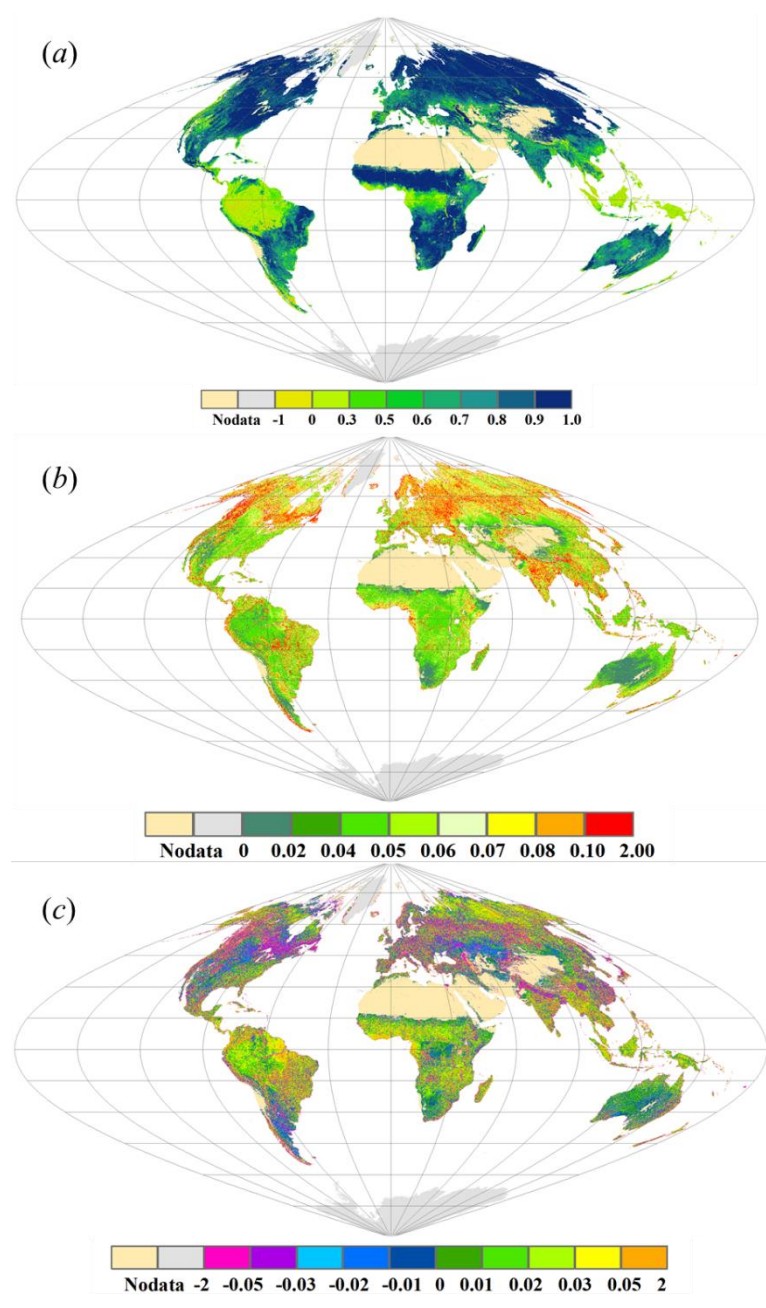


**Figure 14. Spatial distribution of the pixel-by-pixel statistics between the original AVHRR and MODIS data during the overlapping period from 2000 to 2014: (a) *r*, (b) MAD, (c) bias.**





It was found that there is considerable spatial heterogeneity in the global distribution of the spatio-temporal consistency of the fusion results and true MODIS data in the simulated experiments. Some unsatisfactory phenomena were also observed, such as the low $r$ in the area near the equator. This is mainly caused by the cloudy and rainy weather around this area, which results in the vegetation variation not being well monitored by remote sensing compared to the other areas. In order to further explain these phenomena, the original MODIS data were resampled and compared with the AVHRR data in the overlapping time period, and exactly the same statistics were applied, as shown in Fig. 14. Compared with the simulated evaluation results shown in Fig. 4, great similarity can be found for all three indices. These results indicate that the spatial distribution of the fusion accuracy mostly depends on the consistency between the original AVHRR and MODIS data. For the area around the equator, the $r$ between the AVHRR and MODIS datasets was originally poor. This was caused by the rainy and cloudy regional climate is these areas, where the satellite observations were highly impacted by cloud for both satellites (Martins et al., 2018; Schroeder et al., 2008). It is therefore difficult for the NDVI time-series data to represent the real change of surface vegetation, and the temporal variation of the two NDVI products cannot be regarded as the real surface change characteristic. The poor data quality of the optical remote sensing data in these areas has also been widely reported in other research (Gomis-Cebolla et al., 2018; Martins et al., 2018; Tian et al., 2015). As a result, simultaneously impacted by the poor data quality and the differences in transit time and sensor design, inconsistencies exist through all the time periods for the MODIS and AVHRR data. These inconsistencies will certainly be inherited in the fusion results, so that they cannot obtain a similar temporal variation to the true MODIS data with a relatively low $r$. Furthermore, the relatively high MAD and bias in some high-latitude areas in the Northern Hemisphere are also the results of the poor original relationship between the AVHRR and MODIS data. In these regions, there is considerable seasonal heterogeneity in the annual vegetation variation. It was found that the fusion data are superior in the growing season with high NDVI, but are relatively poor in months with low NDVI when the vegetation is withered, as shown in Fig. 8 and Fig. 9. Although the performance of the fusion results is comparatively poor within these regions, it is still within an acceptable range, with the MAD generally less than 0.1 and the bias within ±0.05.

The consistency with the MODIS data is greatly improved after the fusion process, with obviously lower values of MAD and bias in Fig. 4 compared to Fig. 12. The pixels with MAD values greater than 0.1 and bias larger than ±0.05 have been mostly eliminated in the fusion results, whereas such pixels are extensively distributed in the original AVHRR data. In almost all the Southern Hemisphere, the MAD and bias are within an ideal range (MAD<0.04, 0.02<bias<0.02) after the fusion process, but are more than double in the original AVHRR results. Even in the area around the equator, these two indices are significantly improved by the fusion process, with the MAD less than 0.04 and the bias within ±0.01. This indicates that the temporal variation between the fusion data and true MODIS data is unsatisfactory, but the values are quite close. What is more, as shown in Figs. 6 and 7, the fusion results show a very similar spatial distribution to the true MODIS data. As such, even the fusion data cannot be applied to analyze the temporal variation of the vegetation in the area around the equator, which is also impractical for the original AVHRR and MODIS data.



## 5.2 Limitations and uncertainties of the STFLNDVI product

The STFLNDVI product is a new long-term NDVI time series with a spatial resolution of 1 km from the year 1982, which was produced by fusing AVHRR and MODIS datasets. This product represents a good backward extension of the existing MODIS product, and it provides the possibility to form a near 40-year continuous 1-km global NDVI time series, which has not been achieved in previous studies. The STFLNDVI product will provide the necessary data for decades-long regional vegetation studies, which could previously be only conducted at a very coarse spatial resolution (~8 km). More detailed spatial patterns in the STFLNDVI data will be important for the vegetation-related studies at the scale of a province, a small country, or other small regions. In these conditions, the previous datasets could not provide enough information to allow the spatial variation of the vegetation to be recognized, but the STFLNDVI data will make a significant contribution to the identification of vegetation-related ecological spatial patterns and the landscape architecture. Furthermore, the higher spatial resolution will also help to match the footprint of ground site observations, which will contribute a lot to the estimation of various vegetation parameters, such as vegetation productivity, LAI, etc. (Chen, 1999; Reich et al., 1999). Previous studies have proved that remote sensing data with a higher spatial resolution can effectively reduce the uncertainties caused by surface heterogeneity (Guan et al., 2017; Xie and Li, 2020b).

Although it was shown that the STFLNDVI product has good overall accuracies, in both the spatial distribution and the temporal variation, there still are some limitations and uncertainties in this product. Firstly, it was found that the accuracy of the product is mostly dependent on the consistency between the original AVHRR and MODIS data. The fusion process cannot obtain an ideal result in the regions where the relationship between the original data sources is naturally unsatisfactory, which is typically found in the areas around the equator. The use of optical remote sensing data in these areas has always been a difficult problem, and the application of the fusion product is also limited. What is more, although the obtained product is processed to be as close to the MODIS data as possible, to match NASA's subsequent series of satellites, the information about the temporal dynamics during 1982–1999 all comes from the AVHRR data. The fusion process can only combine the spatial information from the referenced pairwise MODIS and AVHRR data (Cheng et al., 2017; Gao et al., 2006). Since many researchers have queried the temporal dynamics of AVHRR data after comparison with other NDVI time series, this query may also be relevant in the STFLNDVI product (Tian et al., 2015; Zhang et al., 2020; Zhang et al., 2017a). However, as there is no other long-term global satellite NDVI dataset available before the 2000s, the information from the AVHRR product is the only resource that can be trusted and applied.

The multi-step framework, including filtering, normalization, and fusion processes, could also have led to cumulative uncertainties in the result. The filtering process was applied to reduce the noise in the MODIS and AVHRR products, to improve the consistency between the two datasets. Although this process has been widely conducted in many vegetation-related studies, it is usually an optional operation. However, it was naturally mandatory for product production to eliminate the uncertainties induced by noise. The normalization and the fusion processing were applied to narrow the sensor differences and improve the spatial resolution. The three processing operations had to be applied in order, so that cumulative





error was inevitable. What is more, although the fusion processes were conducted referring to 12 pairwise data in different

months, considering the seasonal heterogeneity, the overlapping information between the AVHRR and MODIS products was not well utilized. Since the normalization and fusion processes are both based on the overlapping time series of the two basic data sources, they could be aggregated into one processing step to reduce the cumulative error. Future work should be conducted to develop a new processing method to achieve this goal and further improve the accuracy of the obtained NDVI product, by effective use of the information in both the temporal and spatial domains.

**6 Conclusions**

In this study, a new long-term NDVI time series named STFLNDVI was composited and carefully evaluated. The product was produced by fusing AVHRR and MODIS products with three processing steps, in order to respectively eliminate the existing problems of data quality, sensor differences, and resolution limitation. The STFLNDVI product combines the respective advantages of these two data sources, with a spatial resolution of 1 km and a monthly temporal interval since 1982.

The simulated experiments based on the overlapping time span of the AVHRR and MODIS data proved the applicability of the fusion framework, because the fusion results showed similar spatial patterns and temporal variation characteristics to the true MODIS data. Significant positive correlations can be observed in 92.41% of the global land area, and the consistency remains stable in different years and months. This indicates that the temporal distance between the fusion data and the reference data has almost no impact on the accuracy. Furthermore, similar results were also obtained when comparing the

STFLNDVI product with the true GLAP and ANCUS AVHRR observations, with comparable spatial distributions ($r$=0.88) and high temporal consistency ($r$=0.84). The performance of the STFLNDVI product was found to be almost the same as that of MODIS when compared with the AVHRR data, which further proved the similarity between the two data sources. Although some unsatisfactory results were found in the areas near the equator and in high latitudes in the Northern Hemisphere, these were a result of the naturally poor relationship between the original AVHRR and MODIS products. The

consistency was also found to have been greatly improved in these areas after the fusion process, because the pixels with MAD values of greater than 0.1 and bias values of larger than ±0.05 had been mostly eliminated. This product will provide the necessary data for decades-long regional vegetation studies, and will help to advance the understanding of long-term terrestrial ecosystem changes and their interaction with global change.

*Data availability.* This dataset can be downloaded at https://zenodo.org/record/4734593 (Guan et al., 2021). Similar to MODIS data, the dataset is provided in int16 data type with valid value ranges from −2000 to 10000, and the background filled value is −3000. The scale factor of the data is 0.0001.



*Author contributions.* XB.G. and HF.S. designed the experiments. XB.G., YC.W. and D.C. performed the experiments.
XB.G. drafted the manuscript. HF.S., XH.L. and LP.Z. revised the whole manuscript. LW.Y. and XX.L. provided related
data and codes. All authors read and provided suggestions for this manuscript.

*Competing interests.* The authors declare that they have no conflict of interest.

*Acknowledgment.* This research was supported by the National Key Research and Development Program of China
(2017YFA0604402), the National Natural Science Foundation of China (42001371) and the LIESMARS Special Research
Funding. The authors would like to thank the NOAA, NASA, and USGS for providing the necessary datasets. We would
also like to thank Dr. Wu Jingan and Dr. Li Wei for providing the code and packages for the data filtering and spatio-
temporal fusion methods. Special thanks are also given to Prof. Chen Jing M. and all the other researchers who have
provided helpful comments and suggestions.

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
