# Peer review of "Fusing MODIS and AVHRR products to generate a global 1-km continuous NDVI time series covering four decades"

_Earth System Science Data, 2021_

## Author Comment (AC2)

**Response to Comments of Referee #1**

**General Comments:**

**This paper by Guan et al. aims to create a new NDVI dataset through fusion of AVHRR GIMMS 3g and MODIS NDVI dataset. The new dataset, namely, STFNDVI has a monthly temporal resolution and 1km spatial resolution, covering the period of 1981-2000. The authors applied several procedures during this process, including denoise, normalization, spatial-temporal fusion. The algorithm is evaluated during the overlapping period and resultant dataset is compared with LAC and HRPT AVHRR NDVI data which are at higher spatial resolution. Obtaining a global high resolution long-term NDVI dataset can be critical for global change studies. This study presented a first attempt to solve this issue, but from my point of view, it is still a rather premature dataset and have limited value.**

**Response:**

Dear Referee #1,

We are particularly grateful for your deep thoughts and valuable comments. Although the NDVI product presented in our study still has many problems that need further investigation, it is the first attempt to produce the global 1-km long-term NDVI dataset that is helpful for global change studies. According to your comments, we have tried our best to revise the manuscript to make it better, and an item-by-item response follows. In the revised manuscript, we added some experiments to further assess the spatial patterns and data continuity of STFLNDVI, and added more discussions on the uncertainties in the process and results. Furthermore, we are also reproducing the STFLNDVI product by referring to the mean value of MODIS and AVHRR over the overlapping period, in order to reduce the risk caused by the selected one-year reference data.

Once again, we are particularly grateful for your careful reading and constructive comments. Thanks very much for your time.

Best regards,

Ph.D. Xiaobin Guan

**Specific Comments:**

1. **When do we need a high-resolution dataset? The answer seems to be clear, when low resolution dataset cannot provide enough details. This include two major aspects, one is that there is enough spatial heterogeneity at finer resolution, the other is that there is additional information that can only be obtained at this finer resolution. One good example for this second point is the change of land cover, e.g., deforestation or reforestation. Using high resolution data can provide information on when these activities happen and how much do they contribute to the changes of vegetation in addition to the nature factors. Another example is tracking vegetation phenology when multiple biome types co-exist within a pixel, and they respond differently to climate change (see Zhang et al. 2017; Chen et al., 2018). Under these conditions, the sub-pixel spatial patterns within a coarse resolution pixel also changes, but the current algorithm cannot get this information. This is critical issue since getting changes of this sub-pixel patterns is often why people would use high resolution dataset. The current algorithm assumes there is no interannual variations in this sub-pixel variation since the reference sub-pixel spatial pattern is provided by one year of MODIS data. This greatly undermines the value of this high-resolution dataset and the author did not even discuss this aspect. Using long-term high-resolution observations such as Landsat may help solve this issue.**

Response: Thanks for your comments. Indeed, the interannual variations of the sub-pixel spatial patterns within a coarse resolution pixel are essential, especially for the area with land cover changes or significant spatial heterogeneity in biome types. However, it is impossible for all the spatio-temporal methods now to reproduce the detailed spatial patterns in the area with land cover changes, only referring to some limited data with finer resolution. It is still the commonly used way to improve the spatial resolution of long time-series data by referring to some unique pairwise data at different resolutions [1-3]. In this condition, it is still not true to say that there are no interannual variations in the sub-pixel variation. It is because the resample process applied to the coarse data before the spatio-temporal fusion can combining the spatial patterns provided by the surrounded coarse pixels, which have certain interannual variations. As a result, the spatial patterns in the fusion results are enhanced by combing the spatial details from the fine resolution in the reference year and the interannual variations at the coarse resolution in the fusion year. The calculation of weights in the applied spatio-temporal fusion method will force the fusion result to be more similar to the coarse data at the fusion year rather than using the spatial patterns from the fine resolution data at the reference year, in the regions with apparent land cover changes. As a result, there is interannual variations in this sub-pixel variation, but most of them are from the data at coarse resolution. It is afraid that the interannual variations of spatial patterns are hard to be reproduced at fine resolution, because no data can provide the actual spatial variations if not introducing another data at the fusion year.

Using long-term high-resolution observations such as Landsat may help solve this issue, which can provide the interannual variations of fine spatial patterns during the pre-MODIS period. However, it is still a challenge to collaborate with the Landsat images with a long revisit period, that is very different from the AVHRR and MODIS data. The primary purpose of this study is to produce a MODIS-like NDVI time series in the pre-MODIS period, i.e., before 2000s. If we introduce

another data like Landsat observations, it is hard to handle the sensor differences between the three sensors. Furthermore, in the regions without land cover changes, the spatial variations in the fusion results can be reliable, it may induce additional errors if we introduce another data. As a result, using the long-term Landsat observations is exactly a good idea to enhance the interannual variations of the spatial patterns in sub-pixel during the pre-MODIS period, we are also planning to further investigate this issue to obtain better results.

We added some more content on this issue in the section of Discussion to declare the reliability of the spatial variations in the STFLNDVI.

2. **Data quality control. One large difference between GIMMS and MODIS is the data quality control procedure. Since GIMMS does not provide effective quality flag for snow or cloud covered pixels, there can be large differences in early or late growing season in northern high latitudes, as well as the tropical ecosystem, where the authors found large discrepancy during the comparison (Figure 4 and 14). A good practice would be to remove these observations during the per-pixel normalization period based on the MODIS quality flag, and only use the good observations to build the MODIS AVHRR relationship. The author mentioned that they use Whittaker filtering method to reduce noise, however, due to the presence of cloud, snow and aerosols, the anomalies of NDVI are often negatively biased, which cannot be effectively handled by the Whittaker filtering method.**

Response: Thanks for the comments. It is true that GIMMS only provide simple flags for seven types (including good value, NDVI retrieved from spline interpolation (possibly snow), NDVI retrieved from average seasonal profile (possibly snow), and missing data) [4], that is not as effective as MODIS do. The difference and insufficiency in data quality control procedures will certainly induce uncertainties when combing the two data products. As a result, one of the primary purposes of adopting the temporal filtering method is to minimize the uncertainties caused by the different data quality control procedures. In the temporal filtering process, the data quality control information from the two products are utilized respectively in the MODIS and AVHRR data, in order to obtain two high-quality NDVI time series with as little noise as possible. After the temporal filtering process, the relationship between MODIS and AVHRR can be highly improved because most of the unwanted noises in the two time-series have been removed, which is necessary and helpful to the per-pixel normalization. The large discrepancy in the tropical ecosystem is mostly because there are too many points denoted as noises in both the MODIS and AVHRR time series that cannot be well corrected by the temporal filtering process, so both the two data cannot reasonably reflect the actual variation of vegetation in these regions. In this condition, even the observations with poor data quality flags were removed, the relationship between the two sensors in the per-pixel normalization cannot be improved. The comparison for the result of a typical tropical ecosystem (tile h12v09) is shown in Figure 1, it can be found that almost no improvements can be observed in building the MODIS-AVHRR relationship in the results when only using the good observations according to the MODIS quality flag. It would also introduce additional uncertainties when applying the relationship to the data that is denoted as cloud or bad quality, because the relationship is only built by the good observations.

[Figure]

Figure 1. Comparison of the distribution of linear *r* between MODIS and AVHRR data in h12v09 (typical tropical ecosystem): (a) original results of using all points; (b) results of only using good observations according to the MODIS quality flag.

Furthermore, we do not directly use the Whittaker filtering method to reduce noise, but using an improved trend filtering method that combines the respective advantages of the l1 trend filtering and Whittaker filtering method. The Whittaker filtering method can denoises well but usually trend to obtain an over-smoothed result, while the l1 trend filter method can keep the key points well but usually trend to preserve some noises. As a result, the applied trend filtering method combines the advantages of the two methods using the adaptive norms, and the negatively biased anomalies in the NDVI time series are also considered by approaching the upper envelope. It has been proved that ideal NDVI temporal filtering results can be obtained by the method [5].

We added more explanation of the applied trend filtering method in the section of Methods, and some more discussion on the data quality control procedure of the two products in the section of Discussion.

3. **Continuity of the dataset. The authors claim that they generated a high spatial resolution dataset spanning over four decades, I guess that they suggest this new dataset can be used in together with MODIS NDVI. However, using two datasets together may create additional problems. For example, the trend for the first period is provided by AVHRR while the second period is from MODIS. Previous studies have demonstrated that the trend from different sensors can be quite different (e.g., Jiang et al., 2017). There may be additional risks that due to the differences in sensor performance, the NDVI calculated from both sensors may have a non-linear relationship, i.e., the probability density functions (PDF) for each pixel may be different between sensors. This cannot be corrected using the linear regression method as proposed by the authors, but requires additional procedure, e.g., PDF matching. This issue can be easily tested using BFAST or other breakpoint detection algorithms.**

Response: Yes, we suggested that the new dataset STFLNDVI can be used together with MODIS data, to forming a 1-km NDVI time series spanning over four decades. In the future, if the follow-up observation data are consistent with MODIS, they can be directly added to this product to form a spatiotemporal continuous NDVI dataset covering the time period from 1982 to date. Due to the fact that all AVHRR sensors will be retired in the near future, it is imperative to extend NDVI to other sensors to maintain continuity and consistency of this global data set. MODIS is a very suitable selection, because NASA already has detailed plans to maintain the continuity of observations with MODIS in the long future, such as VIIRS planned at least through 2038.

[Figure]

Figure 2. Global distribution of $r$ between MODIS and AVHRR data during the overlapping 2000-2015 time period using a linear model.

As a result, although many studies have declared the inconsistency in the temporal variations between MODIS and AVHRR data, there are still many studies merging the two data for long-term analysis after some corrections [6-8]. As there was no other long-term global satellite NDVI dataset available before the 2000s, the temporal variation information from the AVHRR product is the only resource that can be trusted and applied. Although the linear model seems too simple to correct the difference between AVHRR and MODIS data, our results indicate it can obtain quite ideal results in most of the areas in the world after the temporal filtering method. As shown in Figure 2, the linear $r$ between MODIS and AVHRR data over the overlapping 2000-2015 period is pretty satisfactory, with a value higher than 0.9 in most of the world except for the regions near the equator. Previous studies also showed that although the interpretation of GIMMS NDVI trends in humid areas should be done with certain reservations, it is well-suited with the MODIS for long-term vegetation studies in the non-humid areas. Our result further supports these conclusions that the linear model can well build the relationship between AVHRR and MODIS data in most areas in the world, except for the regions with humid climates, such as the tropical ecosystem near the equator [9]. We have also tried some more methods in the normalization process, such as the moving window correction and many other non-linear functions. However, the results indicated that the linear model has the best performance in building the relationship between the two datasets worldwide.

[Figure]

Figure 3. Comparison of the break points detected by the BFAST method between original AVHRR data and the combined time series using STFLNDVI and MODIS data. The left column are the results for AVHRR time series, and the right column are the results for the combined time series using STFLNDVI and MODIS data. From up to bottom are the results for NDVI at different scales: (a) and (b) are the results for the mean NDVI over the world; (c) and (d) are the mean NDVI for the tile h27v06; and (e) and (f) are the NDVI for one pixel in tile h27v06 with a location of (100,200).

We also further compared the break points in the original AVHRR time series and the time series combining STFLNDVI and MODIS data, based on the BFAST breakpoint detection algorithm. The parameter is set as: minimum segment size h=0.3, season model = 'harmonic' and maximum iteration max.iter=5, both for the two time series. As shown in Figure 3, the left column is the results for the AVHRR time series, and the right column is the results for the combined time series using STFLNDVI and MODIS data. The data at three different scales are used for comparison, respectively are (a) and (b) for the mean NDVI over the world; (c) and (d) for the mean NDVI of tile h27v06; and (e) and (f) for one pixel in tile h27v06 with a location of (100,200). It can be observed that the break points detected in the two long-term NDVI time series are almost the same

for the mean NDVI at the world scale and the single tile scale. No additional break points around the year 2000 can be observed in all the three scales after using the combination of STFLNDVI and MODIS time series. Although the time series combining STFLNDVI and MODIS data shows one additional break point in the time series of the pixel NDVI, it is occurred in 2006 due to the different variations of MODIS NDVI compared to AVHRR data after the 2000s. As a result, the break point of the combined time series using STFLNDVI and MODIS data is not caused by using the two data together. Considering the better data quality of MODIS NDVI, the break point here may be a more reasonable result than using the AVHRR data only.

Some more discussion and explanation on the continuity between STFLNDVI and MODIS data have been added in the section of Results and Discussion in our revised manuscript.

**4. L139, the GIMMS 3g v1 version extends to 2015 December. Did the authors use this newer version?**

Response: Yes, we did use the GIMMS 3g v1 version. we have corrected the coverage period of this product in the revised manuscript, sorry for the mistake.

**5. L175, why 1989-1993, why not longer?**

Response: It is a good question. In this study, there are two different actual AVHRR data used to compare with the produced STFLNDVI time series. One is the global GLAP data covering some period in the 1990s, which is used to assess the global spatial distribution of the STFLNDVI. The other one is the LAC AVHRR data that is only available in the conterminous U.S. and Alaska, and thus is used to assess the temporal variation of the STFLNDVI. Although the LAC AVHRR data is available since the year 1989, it is observed from a series generation of NOAA satellites, i.e., NOAA-11, NOAA-12, and so on. The observations from different satellites have not been calibrated yet, so it would induce significant uncertainties if using the long-time series data from different satellites. As a result, the reason why we only apply the LAC AVHRR data from 1989 to 1993 to assess the temporal variation of the STFLNDVI is to ensure the consistency of the true AVHRR data. In this condition, all the true AVHRR observations are from the NOAA-11 satellite that was launched in 1988 and was placed in standby mode in 1995. Furthermore, we also believe that five years of data is enough to assess the temporal variation of the STFLNDVI, and adding some more years of data will not change the comparison conclusions, since the temporal variation of the applied fusion framework has already been verified in the simulation experiments based on the overlapping time series of MODIS and AVHRR.

We have added more explanation for why only using the LAC AVHRR data from 1989 to 1993 rather than a more extended time series in the section of Data description in our revised version.

**6. L257-258, using one year of data as reference can be risking, for example if drought happens in a savanna ecosystem, the tree-grass difference is greater than normal years, which will affect the spatial patterns at sub-pixel scale.**

Response: Thanks for the constructive comment. It is true that it can be risky only to use one year of data as reference if it suffers from some extreme changes, even though it is commonly used in previous studies [1-3]. Considering there was no reference MODIS data before the 2000s, there are only two methods to solve this problem without introducing additional data. The first one, which would be an ideal solution, is to develop a new method that can utilize all the overlap time series data as references to avoid the impacts of the extreme change in some unique reference data. Thus, the spatial differences between MODIS and AVHRR in every time point can be synthetically referred to, but it is still challenging to develop such a method. Another method is to produce the time series NDVI during the pre-MODIS period referring to the mean data calculated from the overlapping MODIS and AVHRR time series. For example, to fuse the data in January from 1982 to 1999, it can take the mean MODIS and AVHRR data in January during 2000 and 2015 as the reference data. We are reproducing the STFLNDVI product in this way to avoid the problems mentioned above.

**7. L370: I suggest the authors to make comparisons where land cover changes happen during the past decades, for example, "the arc of deforestation" in Amazon, Sahel region in Africa, Northern China, these are research hotspots where high resolution dataset is needed.**

Response: We agree with you that it would be more meaningful to make comparisons in the area where land cover changes have happened during the past decades. As a result, we selected two regions in "the arc of deforestation" in Amazon and Northern China for spatial distribution comparison, and replaced two Patches in Figure 7 in our manuscript that were selected for subset spatial comparison. The locations of the newly selected two patches are shown in Figure 4, denoted as Patch 1 and Patch 3. The comparison results can be found in Figure 5. Similar conclusions can be obtained that the fusion data can clearly provide much more spatial information than the original AVHRR data, with similar patterns to MODIS data. Although it still cannot say that the STFLNDVI can capture the land cover change at a fine scale, it shows the potential of STFLNDVI to enhance the spatial patterns than the original AVHRR data in the area with land cover changes.

[Figure]

Figure 4. Location of the four patches selected for subset spatial comparisons.

**8. L385, to qualitatively analyze the difference, I suggest to add a fourth row showing the difference between the first and third.**

[Figure]

Figure 5. Subset spatial comparisons of the different data. From left to right are Patches 1, 2, 3, and 4 in Fig. 3, respectively; and (a)-(d) are the true MODIS data, (e)-(h) are the original AVHRR data, (i)-(l) are the fusion results, (m)-(p) are the differences between MODIS and AVHRR data, and (i)-(l) are the differences between MODIS and fusion results.

Response: It is a good suggestion that can help to analyze the difference qualitatively. We added another two rows to show the differences between the true MODIS and AVHRR/fusion results. As shown in the last two rows in Figure 5, (m)-(p) are the differences between MODIS and original

AVHRR data, and (i)-(l) are the differences between MODIS and fusion results. Only minor differences can be observed between MODIS and fusion results, with most of the area within the range $\pm 0.05$, only a few pixels showing an absolute difference greater than 0.05. While, the differences between MODIS and AVHRR data are pretty obvious, with lots of pixels showing an absolute difference greater than 0.1. Besides, the distribution of the difference between MODIS and AVHRR is quite similar to the distribution of MODIS data, which means that the spatial patterns in AVHRR are almost missing. In contrast, the distribution of the difference between MODIS and fusion results is almost random, which indicates the fusion results can well reproduce a similar spatial distribution with MODIS.

We have revised Figure 7 in our new manuscript version, and added some related content on the comparison of the difference between MODIS and AVHRR/fusion results.

9. **L400, using this 3D plot does not quantitively provide the information of r since it is difficult to locate the absolute value and. The color scheme also changes for each subplot. You may consider just use 2D plot with year as x-axis, month as y-axis and use color to represent the value.**

Response: Thanks for the comment. We have tried to use a 2D plot to show the intra-annual and seasonal variation of the *r* between the fusion data and true MODIS NDVI. As shown in Figure 6, we found that the 2D plot is not as intuitive as the 3D plot to show the changes of *r* in different months and years. We further corrected the problem in the 3D plot in our original manuscript that the color scheme changes for each subplot, and used the same color scheme for all the subplots. Thus, as shown in Figure 7, the information of *r* can be quantitively provided in the 3D plot, as well as its intra-annual and seasonal changes. We think the corrected 3D plot is a better way the demonstrate the intra-annual and seasonal variation of the *r* between the fusion data and true MODIS NDVI, so we use this strategy in our revised manuscript.

[Figure]

Figure 6. 2D plot of the intra-annual and seasonal variation of the *r* between the fusion data and true MODIS NDVI in the 12 selected tiles. The X-axis is the year, the Y-axis is the month, and the value of *r* is only represented by color.

[Figure]

Figure 7. 3D plot of the intra-annual and seasonal variation of the *r* between the fusion data and true MODIS NDVI in the 12 selected tiles. The X-axis is the year, the Y-axis is the month, and the Z-axis is the value of *r*.

**10. L490, is this r value calculated based on the average value of the 12 months? this should be very high since the spatial details are averaged.**

Response: Yes, the *r* value here is calculated based on the average value from different 12 months, which is only used to show that the fused STFLNDVI shows a similar seasonal variation with the true AVHRR data. It is true that the spatial details are averaged after calculating the mean value in each month, but here we only want to demonstrate that the STFLNDVI do not change the seasonal variation over a specific area after the multi-step processes.

**11. Other minor comments**
**L35, visible->red**
**L59, decades->decadal?**
**L110, "limited attempts" means very few attempts, I guess the authors mean "a few attempts"**
**L130, MOD13A2 has a 16-day temporal resolution, I guess this should be MOD13A3?**
**L179, it is not common to use ecological communities, it usually refers to the group of people who study ecology. I suggest to use ecosystems or biome types.**
**L222, the authors use "prove" several times throughout the manuscript, it is a very strong word that requires rigorous test and derivation. I suggest to use "demonstrate" or "show"**
**L343: why do you need to mention "famous" here?**

**L401, Grammarly incorrect, please rewrite.**

Response: Thanks very much for your careful reading and valuable comments. We have corrected the corresponding places according to the comments. We also have checked through the manuscript to avoid other similar minor issues.

**References:**

[1] Zhou, Junxiong, et al. "Sensitivity of six typical spatiotemporal fusion methods to different influential factors: A comparative study for a normalized difference vegetation index time series reconstruction." Remote Sensing of Environment 252 (2021): 112130.

[2] Boyte, Stephen P., et al. "Fusing MODIS with Landsat 8 data to downscale weekly normalized difference vegetation index estimates for central Great Basin rangelands, USA." GIScience & Remote Sensing 55.3 (2018): 376-399.

[3] Hongtao, Jiang, et al. "Extending the SMAP 9-km soil moisture product using a spatio-temporal fusion model." Remote Sensing of Environment 231 (2019): 111224.

[4] Pinzon, Jorge E., and Compton J. Tucker. "A non-stationary 1981–2012 AVHRR NDVI3g time series." Remote sensing 6.8 (2014): 6929-6960.

[5] Xinxin Liu, et al. " One-Step High-Quality NDVI Time-Series Reconstruction by Joint Modeling of Gradual Vegetation Change and Negatively Biased Atmospheric Contamination." IEEE Transactions on Geoscience and Remote Sensing (2021).

[6] Yang, Wenze, et al. "A novel re-compositing approach to create continuous and consistent cross-sensor/cross-production global NDVI datasets." International Journal of Remote Sensing 42.16 (2021): 6025-6049.

[7] Brown, Molly E., et al. "Neural networks as a tool for constructing continuous NDVI time series from AVHRR and MODIS." International Journal of Remote Sensing 29.24 (2008): 7141-7158.

[8] Mao, Dehua, et al. "Integrating AVHRR and MODIS data to monitor NDVI changes and their relationships with climatic parameters in Northeast China." International Journal of Applied Earth Observation and Geoinformation 18 (2012): 528-536.

[9] Fensholt, Rasmus, et al. "Evaluation of earth observation based long term vegetation trends—Intercomparing NDVI time series trend analysis consistency of Sahel from AVHRR GIMMS, Terra MODIS and SPOT VGT data." Remote sensing of environment 113.9 (2009): 1886-1898.

---

## Author Comment (AC3)

**Response to Comments of Referee #2**

**General Comments:**

Guan et al. generated a new NDVI dataset, STFLNDVI, by merging the data of MODIS NDVI and AVHRR GIMMS 3g. MODIS NDVI product has good data quality and a high spatial resolution but it is available since the year 2000. AVHRR GIMMS 3g product has been provided since 1982 but it has a relatively coarse spatial resolution (1/12 degree) and relatively poor data quality. The authors then performed the temporal filtering, normalization, and spatial-temporal fusing, making a new NDVI dataset of STFLNDVI with 1-km spatial resolution, covering the period of 1982-2015. Furthermore, the authors checked the temporal consistency, spatial stability, and spatial consistency of the new product during the overlapping periods of MODIS, ANCUS NDVI data. This draft was well-written, but I still have some comments on the algorithms used in this analysis, and I think the novelty is insufficient for a paper in ESD.

**Response:**

**Dear Referee #2,**

We are particularly grateful for your careful reading and constructive comments. Although there are still insufficiencies in the STFLNDVI product presented in our study, it is the first attempt to produce the global 1-km long-term NDVI dataset that may be helpful for global change studies. We have taken full consideration of your comments to improve the product and revise the manuscript to make it better. An item-by-item response follows. More experiments are conducted to assess the availability of STFLNDVI on spatial patterns and temporal variations. We are also reproducing the dataset by referring to the mean value calculated from the overlapping MODIS and AVHRR time series, in order to reduce the uncertainties caused by the selected one-year reference data.

Once again, we are particularly grateful for your careful reading and constructive comments. Thanks very much for your time.

Best regards,

**Ph.D. Xiaobin Guan**

**Specific comments:**

1. Doubt on the reliability of the spatial variations at fine resolution. The original AVHRR product at coarse resolution can not provide any spatial variations within 1/12 x 1/12 pixels. The authors claimed that this newly generated NDVI product at 1-km resolution has the information of spatial variations within 1/12 x 1/12 resolution. Such spatial variations for every year are derived from the reference year (2014) of MODIS data. This means that, for STFLNDVI, the spatial variations within 1/12 x 1/12 resolution have no temporal change. This is no realistic, and I think the "high resolution" of STFLNDVI seems like a "pseudo high resolution".

Response: Thanks for your deep thoughts and comments. It is true that the spatio-temporal fusion process was adapted only referring to one year in this study, such as 2000 or 2001 in producing the STFLNDVI and 2014 in the simulated experiments. Although this is the most commonly used method in the spatio-temporal fusion applications, i.e., improve the resolution of time series data only referring to some unique pairwise data [1-3], there may be insufficient spatial variations in the fusion results. The enhanced spatial patterns rely on the selected reference data at finer resolution, and the temporal changes are all from the time series data at poorer resolution. If there were no land cover changes or other extreme changes in the region, the spatial variations in the fusion results would be reliable because the spatial differences between the MODIS and AVHRR data are consistent in the reference year and the fusion year.

As a result, the reliability of the spatial variations in the fusion results may be doubtable mainly for the regions with obvious land cover changes, because the spatial variations within a coarse pixel from the reference year will not be the same as it is in the fusion year. However, we believe all the methods now can not solve this problem, because there are no actual spatial variations at fine-scale that can be obtained in the fusion year. All the useful spatial information in the fusion year is only from the data at coarse resolution, and the spatial variations from fine resolution data at other time is meaningless due to the land cover change. In this condition, our process can only enhance the spatial variations by combing the spatial patterns at fine resolution in the reference year and the information of land cover changes at coarse resolution in the fusion year. Due to the resample process for the coarse data, the spatial variations of fusion results within a coarse resolution is not only decided by the coarse pixel itself, but also impacted by the surrounded coarse pixels. As a result, there do have temporal changes in the spatial variations within the coarse resolution, which relies on the information provided by the coarse resolution data. If there are apparent differences between the two coarse data at fusion year and reference year, which indicates the obvious land cover changes, the calculation of weights in the applied spatio-temporal fusion method will force the fusion result to be more similar to the coarse data at the fusion year rather than using the spatial patterns from the fine resolution data at the reference year. It is afraid that the interannual variations of spatial patterns are hard to be reproduced at fine resolution, because no data can provide the actual spatial variations at the fusion year during the pre-MODIS period.

We added more content on this issue in the section of Discussion to declare the reliability of the spatial variations at fine resolution in the STFLNDVI. Besides, we are also reproducing the STFLNDVI product by referring to the mean data calculated from the overlapping MODIS and AVHRR time series, in order to obtain a more reasonable result and reduce the risk caused by the

**selected one-year reference data.**

2. Doubt on the reliability of the short-term temporal variations. When doing the normalization (section 3.1.2), the authors just used a linear model to make the multi-year mean value and trend of AVHRR data as same as MODIS data (as shown in Fig 3). The interannual variability or temporal variations within the year of STFLNDVI are from the AVHRR data without any correction. The short-term temporal variations of AVHRR aren't always consistent with those of MODIS data, for example in the regions around the equator. Merging two datasets may lead to some artificial variations.

Response: Thanks for the comment. Indeed, the aim of producing the STFLNDVI is to using together with the MODIS data, in order to form a 1-km NDVI time series spanning over four decades. Due to the fact that all AVHRR sensors will be retired in the near future, it is imperative to extend NDVI from other sensors to maintain continuity and consistency of this global data set. As there is no other long-term global satellite NDVI dataset available before the 2000s, the temporal variation information from the AVHRR product is the only resource that can be trusted and applied. Besides, MODIS is a superior instrument to AVHRR in vegetation monitoring, so the temporal variation from MODIS is better to represent the vegetation changes after the 2000s [4,5]. As a result, although many studies have declared the inconsistency in the temporal variations between MODIS and AVHRR data, there are still many studies merging the two data for long-term analysis after some corrections [4-6].

Figure 1. Global distribution of *r* between MODIS and AVHRR data during the overlapping 2000-2015 time period using a linear model.

Since the short-term temporal variations before the 2000s are from the STFLNDVI (i.e., from the AVHRR data only with a linear correction) and the short-term temporal variations after the 2000s are all from the MODIS data, the only problem in the short-term variations may exist around 2000. The applied normalization process is used to correct the short-term variations of AVHRR to be consistent with MODIS. Although the linear model seems quite simple, the results showed in Figure 3 in the manuscript are satisfactory. The results in Figure 1 further supported the satisfactory results of the linear model, which is the linear r between MODIS and AVHRR data over the overlapping 2000-2015 period. A value higher than 0.9 can be found in most of the world except for the region

around the equator. Previous studies also showed that although the interpretation of GIMMS NDVI trends in humid areas should be done with certain reservations, it is well-suited with the MODIS for long-term vegetation studies in the non-humid areas [7]. Our result further supports these conclusions that the linear model can well build the relationship between AVHRR and MODIS data in most areas in the world, except for the regions with humid climates, such as the regions around the equator. In these areas, both the temporal variations in the MODIS and AVHRR data cannot be believed due to the cloudy climate-induced frequent noise in the time series. We have also tried some more methods in the normalization process, such as the moving window correction and many other non-linear functions. However, the results indicated that the linear model has the best performance in building the relationship between the two datasets worldwide.